

# Arctic sea ice drift-strength feedback modelled by NEMO-LIM3.6

David Docquier[1], François Massonnet[1,2], Neil F. Tandon[3], Olivier Lecomte[1], and Thierry Fichefet[1]

[1]Université catholique de Louvain (UCL), Earth and Life Institute (ELI), Georges Lemaître Centre for Earth and Climate Research (TECLIM), Louvain-la-Neuve, Belgium
[2]Barcelona Supercomputing Center - Centro Nacional de Supercomputación (BSC-CNS), Earth Sciences Department, Barcelona, Spain
[3]Environment and Climate Change Canada (ECCC), Climate Research Division, Toronto, Canada

*Correspondence to:* david.docquier@uclouvain.be

**Abstract.** Sea ice cover and thickness have substantially decreased in the Arctic Ocean since the beginning of the satellite era. As a result, sea ice strength has been reduced, allowing more deformation and fracturing and leading to increased sea ice drift speed. The resulting increased sea ice export is thought to further lower sea ice concentration and thickness. We use the global ocean-sea ice NEMO-LIM3.6 model (Nucleus for European Modelling of the Ocean coupled to the Louvain-la-Neuve

sea Ice Model), satellite and buoy observations, as well as reanalysis data over the period from 1979 to 2013 to study this positive feedback for the first time in such detail. Overall, the model agrees well with observations in terms of sea ice extent, concentration and thickness. Although the seasonal cycle of sea ice drift speed is reasonably well reproduced by the model, the recent positive trend in drift speed is weaker than observations in summer. NEMO-LIM3.6 is able to capture the relationships between sea ice drift speed, concentration and thickness in terms of seasonal cycle, with higher drift speed for both lower

concentration and lower thickness, in agreement with observations. Sensitivity experiments are carried out by varying the initial ice strength and show that higher values of ice strength lead to lower sea ice thickness. We demonstrate that higher ice strength results in a more uniform sea ice thickness distribution, leading to lower heat conduction fluxes, which provide lower ice production, and thus lower ice thickness. This shows that the positive feedback between sea ice drift speed and strength is more than just dynamic, more complex than originally thought and that other processes are at play. The methodology proposed

in this analysis provides a benchmark for a further model intercomparison related to the interactions between sea ice drift speed and strength.

## 1 Introduction

The motion or drift of sea ice results from a balance of wind stress, ocean stress, ice internal stress, Coriolis force and ocean surface tilt. Scale analysis shows that the main drivers of drift are the first three terms, and that both Coriolis force and ocean

surface tilt are an order of magnitude smaller (Steele et al., 1997; Leppäranta, 2011). For individual ice floes and for ice fields with low compactness, ice internal stress is generally neglected: sea ice is in free drift. Otherwise, ice internal stress is an important driver of sea ice motion and is a key element of the positive feedback between sea ice drift speed and strength described hereafter. In this paper, we focus on the Arctic Ocean, for which a sufficient network of observations is available.





At large scale, sea ice strength mainly depends on two quantities, namely sea ice concentration (defined as the relative amount of ocean area covered by sea ice) and sea ice thickness. A decrease of concentration or thickness, as observed in the recent decades in the Arctic Ocean (Stroeve et al., 2012; Vaughan et al., 2013; Lindsay and Schweiger, 2015), leads to a reduced ice strength and internal stress, which allows more deformation and fracturing within the ice, hence larger sea ice drift speed (Rampal et al., 2011; Spreen et al., 2011; Kwok et al., 2013). This in turn provides higher export of sea ice out of the Arctic Basin, resulting in lower sea ice concentration and further thinning (Langehaug et al., 2013). On the contrary, an initial increase in concentration or thickness leads to lower drift speed, which finally results in higher concentration or thickness. The mechanism of this positive feedback, which we name 'drift-strength feedback' throughout this paper, is shown in Fig. 1.

A clear increase in Arctic sea ice drift speed has been detected since the 1950s using buoy observations and satellite measurements (Häkkinen et al., 2008; Rampal et al., 2009, 2011; Spreen et al., 2011; Vihma et al., 2012; Kwok et al., 2013; Olason and Notz, 2014). While increased wind speed seems to be the likely cause of the increase in sea ice motion before 1990, the reduced ice strength (caused by reduced thickness and concentration) is the dominant driver since then (Döscher et al., 2014). A direct consequence of higher drift speed is a rise in sea ice export out of the Arctic Basin, which mainly occurs through Fram Strait. However, a distinction needs to be made between area and volume exports: the former is the product of sea ice drift speed, concentration and transect length, while the latter is the product of area export and ice thickness. Several studies show an increase of ice area export at Fram Strait since the late 1970s (Langehaug et al., 2013; Krumpen et al., 2016; Smedsrud et al., 2016), while Kwok et al. (2013) show a small decrease between 1982 and 2009. In terms of volume export, the amount of studies is limited by the relatively low ice thickness temporal coverage at Fram Strait. Spreen et al. (2009) show no significant change in ice volume export between 1990 and 2008. The review from Döscher et al. (2014) summarises this question by stating that no significant long-term trend in sea ice area export is seen due to a balance between increased drift speed and decreased concentration, while volume export slightly falls due to a decreased sea ice thickness. A summary of these studies related to drift speed trend and its cause as well as sea ice export at Fram Strait is provided in Table 1.

Olason and Notz (2014) investigate the interactions between sea ice drift speed, concentration and thickness using satellite and buoy observations. They show that both seasonal and recent long-term changes in sea ice drift are primarily correlated to changes in sea ice concentration and thickness. At seasonal time scales, when sea ice concentration is low (from June to November), drift speed increases with decreasing concentration, while for high concentration (from December to March), drift speed changes are largely driven by changes in thickness (higher drift speed with lower thickness).

An analysis of sea ice outputs coming from the Coupled Model Intercomparison Project 3 (CMIP3) multi-model dataset suggests that thicker and more packed sea ice drifts faster, contrary to what is observed (Rampal et al., 2011). The same study also shows that models with a stronger long-term thinning trend do not exhibit faster drift speed, suggesting that the positive feedback between drift and strength is underestimated in CMIP3 models. According to the authors, this could explain the too low trends in sea ice area, thickness and drift speed.

In the studies previously mentioned, several highlight the drift-strength feedback as an important element of sea ice dynamics but there is no detailed study on its impact on the Arctic sea ice and its relative importance. The main goal of this study is to investigate this feedback using the new version of the Nucleus for European Modelling of the Ocean coupled to the Louvain-





la-Neuve sea Ice Model (NEMO-LIM3.6). In order to perform this analysis, we first study the interactions between drift speed, concentration and thickness by applying the methodology developed by Olason and Notz (2014) to a model. However, good agreement between the modelled and observed drift-strength relationships is not a guarantee that the model is able to capture the drift-strength feedback. Thus, we carry on additional sensitivity experiments where initial ice strength is varied, and we

investigate the impact on the resulting drift speed, concentration and thickness of sea ice. The methodology proposed in this analysis constitutes a benchmark for a further model intercomparison related to the drift-strength feedback and could be used in the analysis of sea ice outputs from the upcoming High Resolution Model Intercomparison Project (HighResMIP) (Haarsma et al., 2016) and CMIP6 Sea-Ice Model Intercomparison Project (SIMIP) (Notz et al., 2016).

In Section 2 we describe the model, the observations as well as the diagnostics and metrics. Then, results from the model

evaluation against different observational and reanalysis datasets are presented (Section 3.1). Section 3.2 details the interactions between drift speed and strength at seasonal time scales. Results from the sensitivity experiments are shown in Section 3.3. These results are discussed in Section 4. Finally, a summary is provided in Section 5.

## 2 Methodology

### 2.1 Model description

The model used in this study is version 3.6 of the global ocean-sea ice coupled model NEMO-LIM. The ocean component NEMO3.6 is a finite difference, hydrostatic, primitive equation model (Madec, 2016). The sea ice component LIM3.6 is a dynamic-thermodynamic model that uses the elastic-viscous-plastic (EVP) rheology on a C-grid and includes an explicit ice thickness distribution (ITD) (Vancoppenolle et al., 2009; Rousset et al., 2015).

The atmospheric forcing is the Drakkar Forcing Set (DFS) 5.2 (Dussin et al., 2016), which is based on the ERA-Interim

atmospheric reanalysis dataset. The model is run on the global tripolar ORCA1 grid (about $1°$ spatial resolution) from January 1958 to December 2015. Model outputs that are used in this study are sea ice concentration, thickness and velocity components over the period 1979-2013. This period is chosen to match the satellite period. Sea ice thickness used in this study is the sea ice volume per grid cell area taking into account open water (named 'sivol' according to SIMIP nomenclature) rather than the actual thickness ('sithick'), since the former is more physical in representing global interactions between sea ice dynamics and

thermodynamics, and this is the variable used in the strength equation (1) below.

Four different simulations are performed in order to test the sensitivity of sea ice drift speed, thickness and concentration to changes in initial sea ice strength $P$. The latter is computed as a function of ice thickness $h$ and concentration $A$ using the formulation of Hibler (1979):

$$P = P^* h^\lambda \exp[-C(1 - A)], \tag{1}$$

where $P^*$ and $C$ are fixed empirical constants set to 20 kN m$^{-2}$ and 20 respectively. An exponent parameter $\lambda$ is introduced into the original Hibler formulation and takes four values corresponding to the four different simulations ($\lambda = 0.5, 1, 1.5, 2$). The experiment with $\lambda = 1$ (original Hibler formulation) is the control simulation that is analysed in Sections 3.1 and 3.2.



The results from the other three experiments are presented in Section 3.3. The experiment with $\lambda = 0.5$ provides lower ice strength than the original formulation ($\lambda = 1$) for a given thickness, while the experiments with $\lambda = 1.5$ and 2 give higher strength. Figure 2 shows how sea ice strength varies as a function of sea ice thickness and concentration for the four $\lambda$ values. The sensitivity experiments presented here allow us to test the representation of the drift-strength feedback in the model.

An increase of $\lambda$ (increase of ice strength) should lead to a decrease in drift speed and a consequent increase in thickness and concentration according to the drift-strength feedback (Fig. 1). If this chain of results is not reproduced by the model, it suggests either that the model can not accurately represent the feedback or that another process is at play.

## 2.2    Observations

In this study, we use several observational datasets for a given variable in order to evaluate model results, following the
recommendations of Notz (2015) and Massonnet et al. (2016).

For sea ice drift speed, the International Arctic Buoy Programme (IABP) C buoy dataset is retrieved from the National Snow and Ice Data Center (NSIDC) (Tschudi et al., 2016). The merged product from Tschudi et al. (2016) suffers from artifacts of the method used to incorporate buoy data (Szanyi et al., 2016), so we prefer using only buoy data rather than the merged product. This dataset provides 12-hourly sea ice velocity vectors derived from buoy positions over the period extending from
January 1979 to May 2015. Ice motion derived from buoys is more accurate than that obtained from satellites (error of less than 1 cm s$^{-1}$ for the average velocity over 24 h according to NSIDC), but the coverage is very limited and the number of buoys and their locations varies from year to year. In addition, buoys have not been placed on ice in the Eastern Arctic. The daily mean sea ice drift speed is computed for each buoy from 12-hourly data.

We also use the low resolution sea ice drift product (OSI-405-b) from the European Organisation for the Exploitation of
Meteorological Satellites Ocean and Sea Ice Satellite Application Facility (EUMETSAT OSI SAF, 2015b; Lavergne et al., 2010). This dataset covers the period from October 2006 to present. No data is available from May to September (inclusive) for the Arctic region due to high atmospheric liquid water content and to ice surface melting. Despite this low temporal coverage compared to other sea ice drift products, the quality of OSI SAF data is superior to other satellite products based on a recent uncertainty estimate (Sumata et al., 2014). This dataset combines satellite measurements of both the brightness
temperature using passive microwave instruments from Special Sensor Microwave/Imager (SSM/I), Special Sensor Microwave Imager Sounder (SSMIS), Advanced Microwave Scanning Radiometer - Earth Observing System (AMSR-E) and Advanced Microwave Scanning Radiometer 2 (AMSR2) and the radar backscatter using Advanced Scatterometer (ASCAT). Sea ice drift vectors are provided every 2 days at a spatial resolution of 62.5 km, and we calculate daily mean drift speed from these data.

Monthly mean ice velocity vectors from the Pan-Arctic Ice-Ocean Modeling and Assimilation System (PIOMAS) are also
used to derive drift speed of sea ice from 1979 to present. PIOMAS is a multi-category thickness and enthalpy distribution sea ice model coupled with the Parallel Ocean Program developed at the Los Alamos National Laboratory (Zhang and Rothrock, 2003). The model assimilates observed sea ice concentration and sea surface temperature and is driven by daily NCEP-NCAR reanalysis surface forcing fields. The mean horizontal resolution in the Arctic is 22 km.



For sea ice concentration, the global reprocessed dataset (OSI-409-a) from OSI SAF (EUMETSAT OSI SAF, 2015a) is used. It covers the period from October 1978 to April 2015 using passive microwave data from Scanning Multichannel Microwave Radiometer (SMMR), SSM/I and SSMIS. The OSI SAF algorithm to retrieve concentration from brightness temperature is a linear combination of the Bootstrap algorithm in frequency mode over open water (Comiso, 1986; Comiso et al., 1997) and

the Bristol algorithm over ice (Smith, 1996). This is one of the best concentration algorithms in terms of precision (standard deviation) according to a recent evaluation (Ivanova et al., 2015). The spatial resolution for this dataset is 10 km. We compute the monthly mean concentration from daily data.

We also retrieve the monthly mean sea ice concentration (1979-2015) computed from the AMSR-E Bootstrap algorithm with daily varying tie-points (Comiso, 2015). This dataset is derived using measurements from SMMR, SSM/I and SSMIS.

Due to orbit inclination, data do not cover the region north of 84.5 °N for SMMR and 87.2 °N for SSM/I and SSMIS. Data are gridded on the SSM/I polar stereographic grid with a 25 km resolution. Largest errors related to this dataset are found in summer when melt is underway.

For sea ice thickness, the gridded data at a spatial resolution of 25 km from ten Ice, Cloud, and land Elevation Satellite (ICESat) campaigns is used (Kwok et al., 2009). The Geoscience Laser Altimeter System (GLAS) is the laser altimeter on

board ICESat that measures sea ice freeboard height, from which sea ice thickness is derived using snow depth and densities of ice, snow and water. The coverage period is limited to the months of October-November and February-March starting in late September 2003 and ending in March 2008. The Kwok et al. (2009) dataset provides the mean sea ice thickness for each of the ten campaigns. The mean absolute uncertainty of sea ice thickness derived from ICESat is 0.21 m in October-November and 0.28 m in February-March (Zygmuntowska et al., 2014).

Since there is no long-term consistent ice thickness data set that covers our study period and due to the high uncertainty inherent to thickness retrievals (Stroeve et al., 2014; Zygmuntowska et al., 2014), we also use monthly mean thickness derived from PIOMAS over the period 1979-2013. Sea ice thickness from PIOMAS agrees well with ICESat data over the region for which submarine data are available (Schweiger et al., 2011).

## 2.3 Diagnostics and metrics

A diagnostic is a measure of one characteristic of a model or an observational dataset, while a metric is a scalar number that compares a diagnostic to some reference (typically observations). In this study, we use both 'standard' diagnostics as well as process-based diagnostics and metrics. The four standard diagnostics that we use are:

- sea ice extent, defined as the total area of ocean with sea ice concentration higher than 0.15

- sea ice concentration, which is the relative amount of ocean area covered by sea ice

- sea ice thickness, defined as the sea ice volume per grid cell area

- sea ice drift speed, which is the velocity of sea ice computed at the daily time scale.



It is important to note that for drift speed, all values given in this study are computed from daily components of sea ice velocity $u_d$ and $v_d$:

$$D_d = \sqrt{u_d^2 + v_d^2}. \tag{2}$$

where $D_d$ is the daily mean drift speed. Monthly mean drift speed computed from daily components of sea ice velocity

is approximately twice higher than monthly mean drift speed computed from monthly components of sea ice velocity with NEMO-LIM3.6 due to higher temporal variability at the daily timescale (Fig. 3). Since we express both modelled and observed drift speeds in km per day (km d$^{-1}$) in this study, using daily components makes more sense than using monthly components. PIOMAS provides only monthly velocity components, so we correct the resulting monthly drift speed by multiplying its value by two based on our previous results (Fig. 3).

From daily values of these four standard diagnostics, we compute monthly means temporally averaged over the period 1979-2013 and monthly trends (i.e. linear regression slopes) over the same period. The maps shown in this paper (Figs. 5, 6, 8, 12) provide monthly means averaged over three consecutive months for winter (January, February, March) and summer (July, August, September) for each grid cell. The plots of mean seasonal cycles (Figs. 4, 7, 10) provide spatial means over the Scientific Ice Expeditions (SCICEX) box from US Navy submarine cruises (Rothrock et al., 2008), which is the domain used

by Olason and Notz (2014) in their study. The SCICEX box is representative of sea ice processes happening in the Central Arctic region and is well covered by observational datasets of sea ice concentration, thickness and drift speed. All maps in this study show the contours of the SCICEX box. We have also tested computing spatial means over a much wider domain taking into account all grid cells between 50 and 90 °N with a concentration threshold of $A \geq 0.1$. Unless specifically mentioned in the text, the default domain that is used in the following sections is the SCICEX box. For computing spatial means, a weight is

given to each grid cell proportional to the grid cell area.

We also use three process-based diagnostics and four metrics that are based on the 'standard' diagnostics described above in order to quantify the ability of NEMO-LIM3.6 to capture the sea ice drift-strength feedback in the Arctic. The use of process-based diagnostics and metrics to try to understand the causes of model biases is highly important (Notz, 2015). They allow us not only to quantify model misfits with respect to sea ice state but also yield insight on the origins of these biases.

The first two process-based diagnostics measure the relationship between sea ice drift speed and concentration and the relationship between drift speed and thickness over the mean seasonal cycle as in Olason and Notz (2014) (Figs. 9, 11). All values are monthly means temporally averaged over 1979-2013 and spatially averaged over the SCICEX box. The novelty compared to Olason and Notz (2014), who only use observations, is that we introduce model results in this analysis. Another difference with Olason and Notz (2014) is that we do not normalise drift speed by wind friction speed since our findings were

not sensitive to such normalisation.

The third process-based diagnostic provides the number of ice-covered grid cells in each sea ice thickness bin range (Fig. 13). In this study, we define 11 equally-spaced classes (0.5 m), except for the last class that includes all grid cells with a thickness above 5 m. This diagnostic provides a way to quantify the heterogeneity of the sea ice thickness distribution.



The four metrics are based on the first two diagnostics and are computed over the mean seasonal cycle (i.e. 12 points). The first two metrics are slope ratios, while the last two metrics are normalised distances. The first metric $s_A$ measures the ratio of the modelled drift-concentration slope to the observed drift-concentration slope. Drift-concentration slopes are computed as the linear regression slopes of the relationships between drift speed and concentration over the mean seasonal cycle. The closer the ratio to 1, the closer the model to the observations. The second metric $s_h$ is similar to the first metric, except that the ratio involves drift-thickness slopes.

The third and fourth metrics quantify the normalised distance (in %) between the model and observations for both the drift-concentration ($\epsilon_A$) and drift-thickness ($\epsilon_h$) relationships respectively:

$$\epsilon_A = \frac{1}{n} \sum_{i=1}^{n} \sqrt{\left|\frac{A_{m,i} - A_{o,i}}{\overline{A_o}}\right|^2 + \left|\frac{D_{m,i} - D_{o,i}}{\overline{D_o}}\right|^2} \times 100, \tag{3}$$

$$\epsilon_h = \frac{1}{n} \sum_{i=1}^{n} \sqrt{\left|\frac{h_{m,i} - h_{o,i}}{\overline{h_o}}\right|^2 + \left|\frac{D_{m,i} - D_{o,i}}{\overline{D_o}}\right|^2} \times 100, \tag{4}$$

where $n$ is the number of months (i.e. 12), the $m$ and $o$ subscripts stand for 'model' and 'observations' respectively, $\overline{A}$, $\overline{h}$ and $\overline{D}$ are the mean concentration, thickness and drift speed (respectively) over the 12 months.

## 3 Results

### 3.1 Model evaluation

The modelled mean seasonal cycle of sea ice extent is in the range of the OSI SAF observational reference. However, a too low extent is simulated in August (bias of 1.6 million km$^2$) (Fig. 4a). This feature is similar to Rousset et al. (2015) who use NEMO-LIM3.6 at 2° resolution.

The modelled mean seasonal cycle of sea ice concentration is very close to OSI SAF observations for all months, except in August when the model underestimates the mean concentration by ∼0.15 (Fig. 4b). This partly explains the too low extent at that time of the year (Fig. 4a). The Bootstrap algorithm provides higher concentration than OSI SAF, especially in summer, but the spatial coverage is limited due to the absence of data close to the North Pole (Section 2.2 and Fig. 5f). When the mean concentration is spatially averaged over the wide domain (north of 50 °N), the model bias is much lower in August but is higher during the rest of the year compared to spatial means computed over the SCICEX box (not shown). A more careful spatial analysis reveals that the model accurately reproduces observed concentration patterns in winter with an overestimation at the ice front (Figs. 5a, 5b, 5c). Furthermore, the model clearly underestimates the observed concentration in the Central Arctic in summer (Figs. 5d, 5e, 5f).

The mean seasonal cycle of sea ice thickness is fairly well reproduced by the model compared to PIOMAS when computing the mean over the SCICEX box with a maximum in May and a minimum in September (Fig. 4c). The model overestimates ice thickness when ice is thicker (January to July) and slightly underestimates it when ice is thinner (August to October). The differences between the model and PIOMAS are much larger when computing the mean over the wide domain (north of 50 °N)





and the seasonal cycle is less consistent, with the model showing peak values from June to August (not shown). We do not show the seasonal cycle of ice thickness from ICESat due to the sparse temporal coverage of these satellite data. However, the model reproduces well the spatial distribution of ICESat thickness with thicker ice north of Greenland and in the Canadian archipelago. The model slightly overestimates the sea ice thickness provided by ICESat in February, March and April (mean

bias of 0.07 m), and underestimates it in October and November (mean bias of -0.65 m).

Compared to IABP buoy observations, the model overestimates sea ice drift speed for all months with higher differences from December to March and for June and July (Fig. 4d). The too strong intensity of the modelled sea ice velocity was already shown with NEMO3.1-LIM2 (Dupont et al., 2015). However, the model captures the seasonality of drift speed with higher values in summer, when concentration and thickness are the lowest, and lower values in winter, when concentration and

thickness are high. The minimum modelled drift speed lags the observed minimum by one month (March) and the maximum occurs two months earlier than the observed maximum (September). The recalibrated drift speed in PIOMAS reanalysis is in the range of values given by buoy observations and NEMO-LIM3.6 but there is no clear minimum and the maximum occurs in October. When the mean drift speed is averaged over the wide domain (north of 50 °N), the model bias (NEMO-LIM3.6) is much higher compared to IABP (not shown). This is partly caused by too high modelled drift speed in the straits (particularly

at Fram Strait, Fig. 6). Sea ice drift speed from OSI SAF is not shown in Fig. 4d due to the absence of data in summer, but a spatial analysis shows that NEMO-LIM3.6 overestimates these satellite observations during the rest of the year (mean bias of 0.91 km d$^{-1}$; Fig. 6c). The main patterns of sea ice circulation, i.e. Beaufort Gyre and Transpolar Drift, are reasonably well represented by the model (Figs. 6a and 6d).

Monthly mean trends in sea ice extent are clearly negative all year long in the model and OSI SAF observations, especially

in summer, but the model underestimates these trends (Fig. 7a). For sea ice concentration trends averaged over the SCICEX box, the agreement between the model and observations is much better but the model overestimates the negative trend in August and September (Fig. 7b). Furthermore, monthly mean thickness trends lie within the PIOMAS range but the seasonal cycle amplitude is lower (Fig. 7c). Finally, monthly mean trends in drift speed simulated by the model agree well with IABP from November to June but are clearly out of range from July to October, with a negative modelled trend in August (Fig. 7d).

However, modelled trends are not significant at the 5% level from July to September and agree well with PIOMAS at that time of the year. It is also important to reiterate that these results are valid for means computed over the SCICEX box and do not take into account grid cells outside of this box.

Figure 8 shows spatial variations of modelled monthly mean trends in sea ice drift speed and concentration. Interestingly, the summer trend in drift speed is positive in the western part of the SCICEX box and negative in the eastern part (Fig. 8b).

This effect is particularly enhanced in August (Fig. 8c), which explains the negative trend in drift speed when averaged over the whole SCICEX box (Fig. 7d). The cause of the negative drift speed trend in August in the eastern Central Arctic is due to the removal of sea ice in this region after 1979, as shown by the modelled trend in sea ice concentration in August (Fig. 8f). Therefore, the model bias in summer drift speed trend is probably due to the absence of summer sea ice in the eastern Central Arctic.



## 3.2 How does sea ice drift relate to ice strength?

The feedback between sea ice drift speed and strength can be quantified via the relationships between drift speed and concentration on the one hand and drift speed and thickness on the other hand. In this study, we analyse these relationships in terms of mean seasonal cycles. The linear relationship between drift speed and concentration is clear for both the model and the observations when concentration is relatively low (i.e. in summer): the lower the concentration, the higher the drift speed, with significant slopes at the 5% level (Fig. 9a). However, this relationship does not hold with the drift speed from PIOMAS (the slope is not significant at the 5% level). The modelled drift-concentration slope is weaker than the observed ones (using IABP for drift speed) mainly due to too low mean modelled sea ice concentration in August and too high mean modelled drift speed in March and April. The modelled drift-concentration relationship is in better agreement with the observed IABP/OSI SAF pair (slope ratio $s_A = 0.5$ and normalised distance $\epsilon_A = 3\%$) compared to IABP/Bootstrap ($s_A = 0.3$ and $\epsilon_A = 3.5\%$).

The relationship between drift speed and thickness shows a similar general pattern as the drift-concentration relationship with higher drift speed for lower thickness, with significant slopes at the 5% level (Fig. 9b). However, the drift-thickness relationship is more complex with a hysteresis loop for both NEMO-LIM3.6 and the IABP/PIOMAS pair. From May to August, during the melting season, sea ice thickness decreases and drift speed increases. Then, from September to March, drift speed decreases and thickness increases. The behaviour is slightly different between the model and observations, with a clearer linear relationship from May to August for observations. For both the model and observations, for a given thickness, the drift speed can take two values depending on the season: a high value in summer when sea ice melts and a low value in winter when sea ice forms and grows. PIOMAS drift speed does not reproduce this loop, although the slope ratio and normalised distance between the model and the PIOMAS/PIOMAS pair ($s_h = 1.1$, $\epsilon_h = 3.3\%$) are better than between the model and the IABP/PIOMAS pair ($s_h = 0.5$, $\epsilon_h = 3.5\%$).

Similar scatter plots are produced for the monthly mean trends and show larger differences than monthly mean values between the model and the observed pairs using IABP for drift speed (Figs. 9c and 9d). The positive slope of the modelled relationships is largely driven by low trends in summer, while the observations (with IABP for drift speed) show a decrease in drift speed trend with more positive trends in concentration and thickness. However, the low trends in modelled drift speed in summer are due more to a progressive removal of sea ice rather than an actual decrease in sea ice motion in past years as previously shown (Section 3.1 and Fig. 8). Modelled trends in drift speed and concentration in winter are very close to each other, while the spread is larger for observations. The agreement is better between the model and the pair using PIOMAS for drift speed, but the drift-concentration and drift-thickness relationships (in trends) are likely not reliable for PIOMAS based on previous results (Figs. 9a and 9b).

The analysis of drift-concentration and drift-thickness relationships demonstrates that NEMO-LIM3.6 captures reasonably well the drift-strength feedback at seasonal timescales. The trend relationships are less convincing but two months (August and September) particularly dominate the signal due to a progressive removal of sea ice in the past years within the domain of study (SCICEX box). The use of a wide domain (all grid cells north of 50 °N) provides results for drift-concentration and drift-thickness relationships that clearly diverge compared to observations due to the inclusion of coastal grid cells that have





high model biases (not shown). Computing mean concentration, thickness and drift speed over the Central Arctic (SCICEX box in our analysis) provides values that are much more consistent with observations and more representative of Arctic conditions.

### 3.3  Sensitivity to changes in ice strength

The previous scatter plots (Fig. 9) are valuable to provide insight regarding the interactions between sea ice dynamics and state
variables (concentration and thickness), but they do not quantify the impact of a change in sea ice state on the dynamics of the system. In order to do this, we introduce a $\lambda$ parameter in Eq. (1), as described in Section 2.1 and in Fig. 2, and vary it between 0.5 and 2 ($\lambda = 1$ is the original value in Hibler (1979)) to test the sensitivity of an initial change in ice strength on the resulting sea ice drift speed, concentration and thickness.

Varying this exponent parameter leads to tiny differences in mean sea ice concentration (not shown) but has a significant
impact on sea ice thickness and drift speed (Fig. 10). For higher values of $\lambda$, i.e. larger ice strength for a given thickness, the mean sea ice thickness is lower throughout the whole year (Fig. 10a) and the mean drift speed is lower in winter and spring and higher during summer and fall (Fig. 10b). The amplitude of seasonal cycles in thickness and drift speed is lower with all $\lambda$ values when using the wide domain (north of 50 °N) but the general trend is similar (not shown). Lower sea ice thickness with higher ice strength (Fig. 10a) is counter-intuitive with respect to the drift-strength feedback (Fig. 1). This is further discussed
later.

Higher sea ice drift speed with higher ice strength from July to September (Fig. 10b) is also in contradiction with the expectations from the positive drift-strength feedback. It probably stems from the fact that for higher strength, ice thickness is lower at the beginning of the summer (Fig. 10a), leading to higher drift speed values. The modelled drift speed is closer to observations with $\lambda = 2$ from September to March, $\lambda = 1.5$ in April and May, $\lambda = 0.5$ in August, and all the modelled curves
are out of the observed range in June and July. Therefore, there is no single $\lambda$ value that provides a best estimate for drift speed.

Figure 11 shows the drift-concentration and drift-thickness relationships for the model with different initial $\lambda$ values (only $\lambda = 1$ and $\lambda = 2$ are shown for the drift-thickness relationship) as well as observations in terms of mean seasonal cycle averaged over the period 1979-2013 and over the SCICEX box. All model simulations but one ($\lambda = 0.5$) provide coherent relationships with significant slopes at the 5 % level, i.e. a decreasing drift speed with increasing concentration and thickness as well as a
hysteresis loop for the drift-thickness relationship. For the drift-concentration relationship (Fig. 11a), the $\lambda = 1.5$ curve is closer to observations in terms of slope ratio ($s_A = 0.7$) and normalised distance ($\epsilon_A = 2.3\%$). For the drift-thickness relationship (Fig. 11b), the slope ratio is the highest with $\lambda = 2$ ($s_h = 0.8$) and the normalised distance is the smallest with $\lambda = 1.5$ ($\epsilon_h = 2.7\%$). It is also clearly apparent from Fig. 11b that a higher $\lambda$ parameter value leads to lower thickness and a higher amplitude of the seasonal cycle of drift speed.
Increasing ice strength via the $\lambda$ parameter leads to lower thickness values (Fig. 10a). A more careful spatial analysis allows us to see that this lower ice thickness appears everywhere in the Arctic and during all months of the year, with the most visible differences occurring north of Greenland where ice is the thickest (Fig. 12). This counter-intuitive result (given the positive drift-strength feedback) is due to the more uniform sea ice thickness distribution resulting from a higher strength. Figure 13 shows the distribution of ice-covered grid cells in each thickness bin for both winter and summer months: the higher the ice



strength ($\lambda$), the higher the number of grid cells in the modal class (2.5-3 m in winter and 1.5-2.5 m in summer), and the more uniform (i.e. peaked) the thickness distribution. This higher uniformity in thickness distribution results in smaller heat loss to the atmosphere compared to a more heterogeneous thickness distribution, which leads to lower sea ice production in winter. This is confirmed by the analysis of sea ice thickness in the beginning of model simulations (i.e. in 1958), when

thickness differences between the four simulations are caused only by strength differences (and not by the mean state). Thus, thickness gets lower with higher initial ice strength. The best $\lambda$ option related to this diagnostic lies between 1.5 and 2 when compared to PIOMAS thickness. Therefore, we conclude that the negative thermodynamic feedback between sea ice thickness and heterogeneity competes with (and dominates) the positive drift-strength feedback in these sensitivity experiments.

## 4 Discussion

### 4.1 Novelties of the present study

The main novelty of the present study is the analysis of the positive feedback between Arctic sea ice drift and sea ice state (concentration and thickness) in great detail. Rampal et al. (2011) mention this feedback as an important element of Arctic sea ice processes and other studies analyse interactions between some elements of the system (Spreen et al., 2011; Kwok et al., 2013; Langehaug et al., 2013; Olason and Notz, 2014) but none of these studies quantifies the magnitude of the drift-strength

feedback in detail. Here we address this issue using NEMO-LIM3.6 at 1° resolution with several process-based diagnostics and metrics as well as sensitivity experiments with different values of initial sea ice strength.

  The drift-concentration and drift-thickness diagnostics and metrics used here are based on the work from Olason and Notz (2014). They analyse the interactions between the three variables using different observational datasets within the SCICEX box. At the seasonal timescale, they find that sea ice concentration controls sea ice drift speed in summer (when concentration

is relatively low) and sea ice thickness is the main driver in winter (when concentration is relatively high). In our analysis, we also find that drift speed is anti-correlated to concentration during summer months in the same SCICEX box (Fig. 9a). However, we do find that drift speed is anti-correlated to thickness not only during winter months but also during summer (with a clearer relationship for the observations compared to the model, Fig. 9b). In our study, we do not normalise sea ice drift speed by wind friction speed as in Olason and Notz (2014) because the same relationship is found with and without normalisation. In sum,

we find that on seasonal timescales drift speed is controlled by both thickness and concentration.

  A previous study demonstrated the impact of increased ice strength using a coupled ice-ocean model to account for large-scale effects (Häkkinen and Mellor, 1992). They use both the classical Hibler parameterisation for ice strength as well as a square dependence of ice strength on thickness for first-year sea ice to account for large-scale effects following Overland and Pease (1988) and compare both approaches. They show that increased ice strength leads to thicker ice, which is different from

what we find in this study (increased strength leads to thinner ice). However, Häkkinen and Mellor (1992) do not use an ITD scheme and the sea ice extent is better simulated by the Hibler parameterisation compared to Overland and Pease (1988). Since the presence of an ITD scheme in a sea ice model modifies the relationship between thickness and strength (Holland et al.,



2006), this may explain the discrepancies between our results and Häkkinen and Mellor (1992). Some sea ice models that include an ITD scheme use $P$ scaling as $h^{3/2}$ so that the sensitivity of strength to thickness is higher (Lipscomb et al., 2007).

Our study can also be used to identify what is the best option for $\lambda$ in the Hibler equation (1) by comparing the different simulations to observations. However, the best match is strongly dependent upon the month of the year. For sea ice thickness, the amplitude of the modelled seasonal cycle is higher than the observed one, so that all $\lambda$ values ranging from 0.5 to 2 could be used depending on the month ($\lambda = 1.5$ for January-March and June-July, $\lambda = 2$ for April-May, $\lambda = 1$ for August and November-December, and $\lambda = 0.5$ for September-October) (Fig. 10a). For drift speed, the highest values ($\lambda = 1.5$ and 2) better match observations in winter, and it is difficult to find a best option in summer (Fig. 10b). Therefore, we do not particularly recommend a given value of $\lambda$ but rather recommend to exclude the lowest value ($\lambda = 0.5$) from the possibilities due to a weaker representation of the drift-concentration and drift-thickness relationships (Fig. 11). This shows the limitations of the parameterisation from Hibler (1979) and could support the use of new sea ice rheologies, such as the elasto-brittle rheology (Girard et al., 2009).

## 4.2 Complexity of the drift-strength feedback

The chain of causality involved in the positive feedback between sea ice drift and strength, as initially presented and understood from the literature, is represented by blue boxes and arrows in Fig. 1. An initial decrease in sea ice concentration or thickness leads to a decrease of sea ice strength and internal stress. This results in larger deformation and enhances drift speed and the subsequent export of sea ice out of the Arctic Basin. All this finally leads to further decreases in sea ice concentration and thickness. However, this study shows that the feedback loop is more complex for two main reasons.

The first complication is the fact that a change in export of sea ice is not only a cause but also a consequence of changes in concentration and thickness. The volume export is the product of sea ice drift speed, concentration and thickness. Therefore, a decrease of concentration or thickness will both emphasize the positive feedback by increasing drift speed and the resulting export of sea ice, and reduce its magnitude by directly decreasing export of sea ice (red arrows in Fig. 1). This will balance the resulting sea ice export.

We compute sea ice export at Fram Strait from sea ice drift speed, thickness (which takes into account concentration, since our thickness is the sea ice volume per area) and transect length at two different latitudes (76°N and 80°N) following Spreen et al. (2009). From 1979 to 2013, the volume flux modelled by NEMO-LIM3.6 decreases with a negative trend of 1.8 km$^3$ d$^{-1}$ decade$^{-1}$ in the control simulation ($\lambda = 1$) but interannual variations are very large, especially at the southern transect (76°N). Increasing ice strength (by increasing $\lambda$) results in a decreasing volume flux at Fram Strait mainly due to the lower thickness. Therefore, in our model the direct effect of decreasing concentration and thickness is more important than the impact of increasing drift speed on the export at Fram Strait.

The second difficulty is the addition of another negative thermodynamic feedback linked to sea ice thickness and strength, which explains the decreasing sea ice thickness for an initially increasing ice strength (Section 3.3). A higher strength results in a less heterogeneous sea ice thickness distribution during all months of the year (Fig. 13), which leads to smaller heat losses to the atmosphere, smaller production of sea ice in winter, and thus thinner ice. We update our initial drift-strength




feedback diagram by taking into account this negative feedback (yellow boxes in Fig. 1). These findings are consistent with results obtained with NEMO-LIM3.6 using two different spatial resolutions: the lower resolution also results in a more uniform thickness distribution, which leads to lower ice thickness. Thus, isolating the drift-strength feedback with a set of sensitivity experiments is difficult.

An additional element that has not been studied here and could increase the complexity of the feedback is the interaction between the ocean-sea ice system and the atmosphere. In this analysis, we use an ocean-sea ice model forced by atmospheric reanalysis. A full coupling with the atmosphere could provide different results regarding drift-strength interactions. For example, Juricke and Jung (2014) find that the implementation of a stochastic sea ice strength parameterisation leads to different responses in both the coupled ECHAM6-FESOM model and the FESOM model forced by atmospheric fluxes generated by

the coupled model. In the uncoupled simulation, the Arctic sea ice volume increases compared to a reference run without parameterisation, while the volume remains largely unchanged in the coupled simulation. This suggests that a negative atmospheric feedback explains the differences between both coupled and uncoupled modes. Therefore, care needs to be taken when extrapolating results from forced simulations to coupled models. Specifically for our study, the effect of coupling on the drift-strength feedback could be assessed by comparing our results to coupled simulations using NEMO-LIM3.6 (e.g. EC-Earth in

the framework of CMIP6).

### 4.3   Impact of domain choice

In this study, we compute spatial means both over the SCICEX box, roughly corresponding to the Central Arctic, and over a wider domain encompassing all grid cells north of $50°$N with a concentration threshold ($A \geq 0.15$). For physical reasons and figure readability, the former domain is preferred in our study. Furthermore, from the results obtained and already discussed in

Section 3, it is clear that the latter domain is too vast to give a good agreement with observations. Particularly, using a too wide domain such as our second domain includes the large model biases occurring in the vicinity of the ice front and of straits (e.g. model overestimation of drift speed at Fram Strait). A decomposition of the Arctic Ocean into sub-regions such as in Koenigk et al. (2016) would be a better alternative. However, the comparison between our central and wide domains demonstrates the impact of domain choice, with generally higher skills when using the central domain.

### 4.4   How can this methodology help in future model intercomparisons?

The methodology proposed in this analysis, particularly the process-based diagnostics and metrics, can be used to assess the performance of other models against observational datasets, which will be an important component of SIMIP (Notz et al., 2016). It can be extended to models being forced by atmospheric reanalysis (such as the model used here) as well as fully coupled models, in order to provide a benchmark for further model intercomparison. Process-based metrics, such as slope

ratios and normalised distances, help identify which models fall within the observational range. However, the use of a single number always needs to be put in perspective with the broader picture. For example, a model with a good slope ratio might poorly represent drift-thickness hysteresis (e.g. the PIOMAS/PIOMAS pair in Fig. 9).



In this study, we only focus on one model resolution. Although some preliminary results show that a higher spatial reso-
lution with the same model provides a higher sea ice thickness, there is a need for a deeper analysis of the impact of model
resolution. The methodology proposed fits quite well into the framework of the EU Horizon 2020 PRIMAVERA project
(https://www.primavera-h2020.eu/), which aims at evaluating the effect of high resolution in global climate models.

Finally, this process-based analysis provides an alternative to classic model evaluations that only look at sea ice extent and
thickness. It systematically highlights the links between sea ice dynamic and thermodynamic processes. Evaluating new sea
ice rheologies using this methodology will provide a stronger test of model performance.

## 5   Conclusions

**This study first shows that the global ocean-sea ice NEMO-LIM3.6 model is able to reproduce the observed sea ice**
**extent, concentration, thickness and drift speed reasonably well over the historical period (from 1979 to 2013).** Monthly
mean trends in concentration, thickness and drift speed are also in the observational range, except for the trend in drift speed
during summer. We show that this model bias is linked to the removal of sea ice in the eastern Central Arctic rather than an
actual decrease of drift speed.

    **The interactions between sea ice drift speed and strength are well represented through the relationships between drift**
**speed and concentration on the one hand, and drift speed and thickness on the other hand.** In particular, the increasing
drift speed with lower concentration and thickness is reproduced by the model. The drift-thickness relationship is marked by
a hysteresis loop: two drift speed values are possible for a given thickness depending on the season, with a higher sea ice drift
speed during the melting season and a lower value during the growing season. When considering the relationships between the
trends in drift speed, concentration and thickness, the spread between the model and observations is higher mainly due to too
low summer trend in modelled drift speed.

    **Sensitivity experiments provide counter-intuitive results related to the positive drift-strength feedback.** With higher
initial sea ice strength, we would expect higher ice thickness. However, the opposite happens due to a higher uniformity of
sea ice thickness distribution with a higher initial ice strength, leading to lower conduction fluxes and smaller ice production.
This negative thermodynamic feedback is therefore competing with the positive drift-strength feedback and dominates it in the
context of these experiments. These experiments also show that no single value of thickness exponent is the best option for
reproducing the observed drift speed and thickness, but the best option depends on the variable and on the month of the year.
We do not recommend an ad hoc variation of the Hibler parameterisation, and we instead suggest that other rheologies might
be more appropriate.

    **Finally, this study shows that the interactions between sea ice dynamics and state are more complex than previously**
**thought.** The diagnostics and metrics proposed in this study that relate drift speed to concentration and thickness are nec-
essary conditions for representing the drift-strength feedback, but they are not sufficient. Sensitivity experiments in which
sea ice strength is varied are also essential for validating the feedback. While NEMO-LIM3.6 correctly represents the drift-
concentration and drift-thickness relationships, sensitivity experiments show that processes other than the drift-strength feed-



back are more important in driving sea ice thickness and drift speed responses. It is thus hard to isolate the drift-strength feedback from other processes. In this analysis, we use one resolution of one model with an atmospherically forced mode. A multi-model assessment using different model resolutions, e.g. in the framework of the EU Horizon 2020 PRIMAVERA project, will provide further insight into the interactions between sea ice dynamics and state.

## 6   Code availability

All codes for computing and plotting the results of this article are written in Python programming language and are available upon request.

*Author contributions.*   DD, FM and TF designed the experimental study. FM performed the model simulations. DD collected the observational datasets, developed the diagnostics and metrics, analysed the results and produced the figures. NFT and OL provided substantial feedback to the analysis. DD prepared the manuscript with contributions from FM, NFT, OL and TF.

*Competing interests.*   The authors declare that they have no conflict of interest.

*Acknowledgements.*   DD works on the PRIMAVERA project (PRocess-based climate sIMulation: AdVances in high-resolution modelling and European climate Risk Assessment), which is funded by the European Commission's Horizon 2020 programme, grant agreement no. 641727. FM is funded by the Belgian Fonds National de la Recherche Scientifique (FNRS) and was funded by the Ministerio de Economía, Indistria y Competitividad (MINECO). OL is a research assistant within the Belgian FNRS. NFT is supported by the Canadian Sea Ice and Snow Evolution (CanSISE) Network. Computational resources have been provided by the Consortium des Équipements de Calcul Intensif (CÉCI), funded by the Fonds de la Recherche Scientifique de Belgique (F.R.S.-FNRS) under grant no. 2.5020.11. We would like to thank H. Goosse, M. Vancoppenolle, A. Barthélemy, J. Raulier and V. Dansereau for their very helpful comments regarding this study. We also acknowledge P.-Y. Barriat for his help in using computing resources at UCL and D. François for his advice in improving Python scripts.



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



**Figure 1.** Diagram of the positive sea ice drift-strength feedback (blue boxes) together with the negative sea ice thickness-heterogeneity feedback (yellow boxes). Ice strength and thickness participate in both feedbacks. Arrows show the variable dependencies, with the beginning and end sides showing causes and effects, respectively. A plus sign means that an increase (decrease) in the cause leads to an increase (decrease) in the consequence. A minus sign means that an increase (decrease) in the cause leads to a decrease (increase) in the consequence. Red arrows denote an additional effect that dampens the positive drift-strength feedback. Equation (1) is shown under the 'ice strength' box to recall the relationship between ice strength $P$, thickness $h$ and concentration $A$.





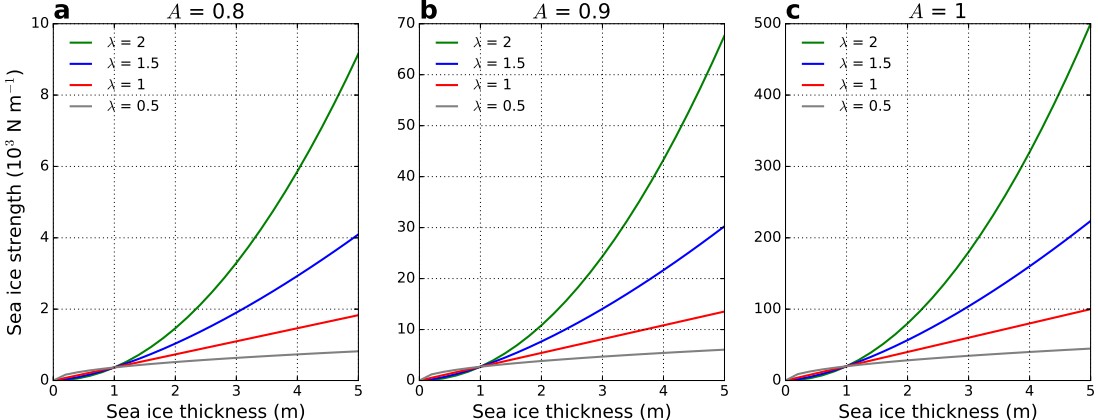

**Figure 2.** Sea ice strength as a function of sea ice thickness computed from Hibler (1979) equation (1) for four given values of $\lambda$. The three panels correspond to three different values of sea ice concentration: (a) $A = 0.8$, (b) $A = 0.9$, (c) $A = 1$. Note that the y axes are different in the three panels.

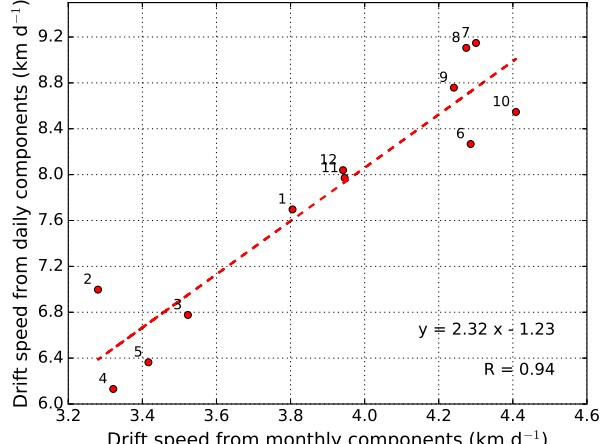

**Figure 3.** Scatter plot of modelled (NEMO-LIM3.6) monthly mean sea ice drift speed computed from daily components against drift speed based on monthly components. Data are temporally averaged over the period 1979-2013 and spatially averaged over the SCICEX box. Numbers denote months. The dashed line represents the linear regression. The equation of the linear regression and the Pearson correlation coefficient between both variables are shown in the lower right corner.





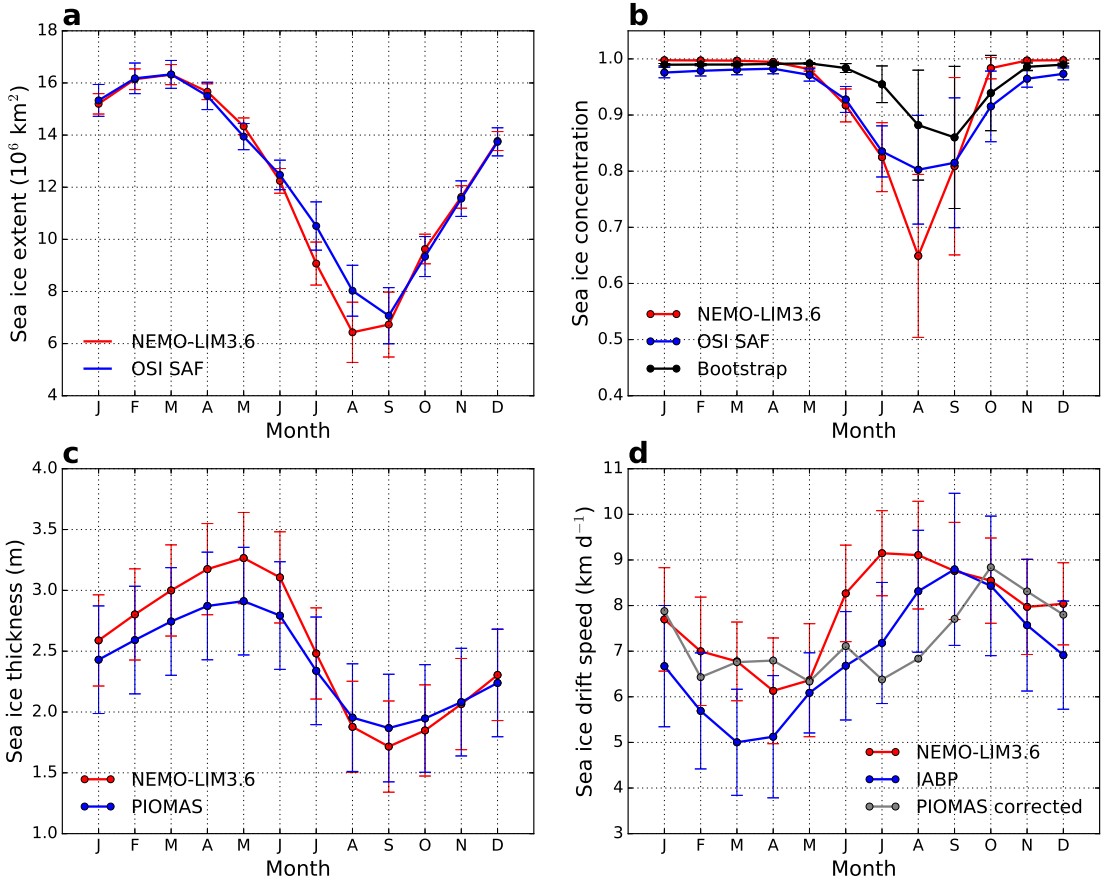

**Figure 4.** Modelled (NEMO-LIM3.6) and observed monthly mean seasonal cycles of Arctic sea ice (a) extent (total area of grid cells where concentration is higher than 0.15), (b) concentration, (c) thickness (sea ice volume per area) and (d) drift speed averaged over the period 1979-2013. The spatial mean over the SCICEX box is represented in (b), (c) and (d). Sources for observations: OSI SAF satellite data for extent and concentration, Bootstrap satellite data for concentration, PIOMAS reanalysis for thickness and drift speed, and IABP buoys for drift speed. PIOMAS drift speed data are computed from monthly velocity components and are multiplied by two to be comparable to other drift data. Error bars show the temporal standard deviation of monthly values.





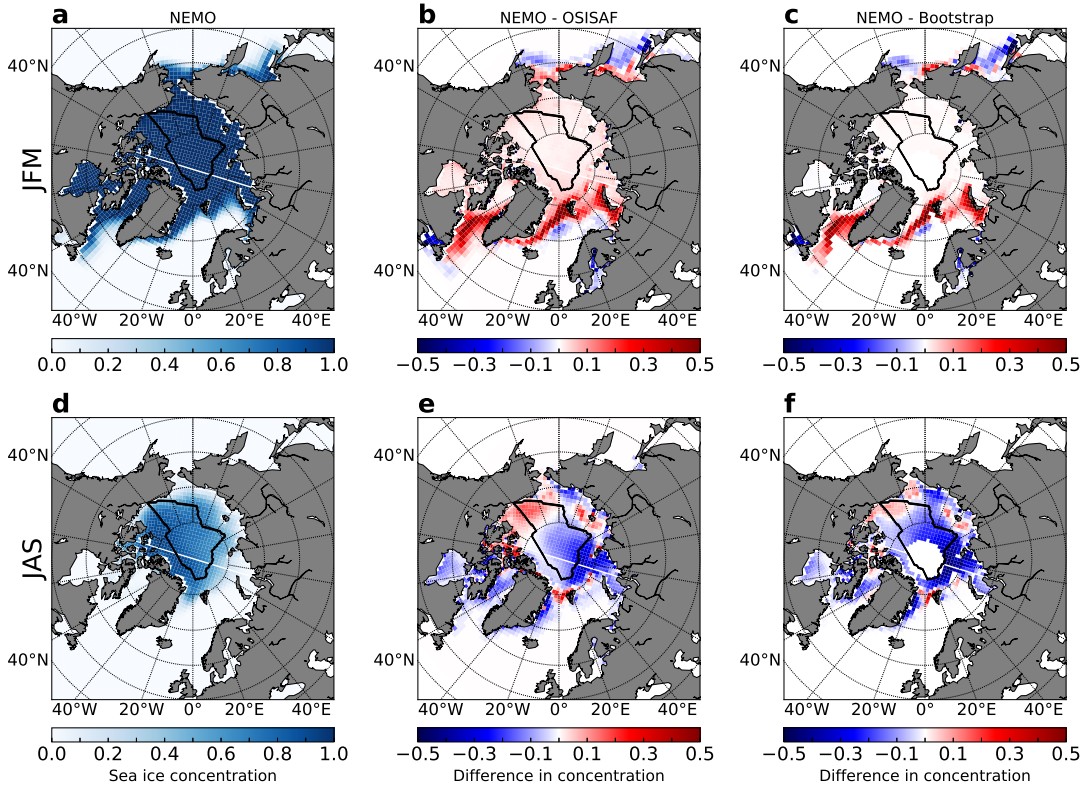

**Figure 5.** (a) Modelled (NEMO-LIM3.6) mean Arctic sea ice concentration averaged over the winter months (JFM, i.e. January-February-March) of the period 1979-2013. (b) Difference in concentration between NEMO-LIM3.6 and OSI SAF averaged over the winter months of the period 1979-2013. (c) Difference in concentration between NEMO-LIM3.6 and Bootstrap averaged over the winter months of the period 1979-2013. (d), (e), (f) Same as (a), (b), (c) respectively for the summer months (JAS, i.e. July-August-September). The black polygon is the contour of the SCICEX box.





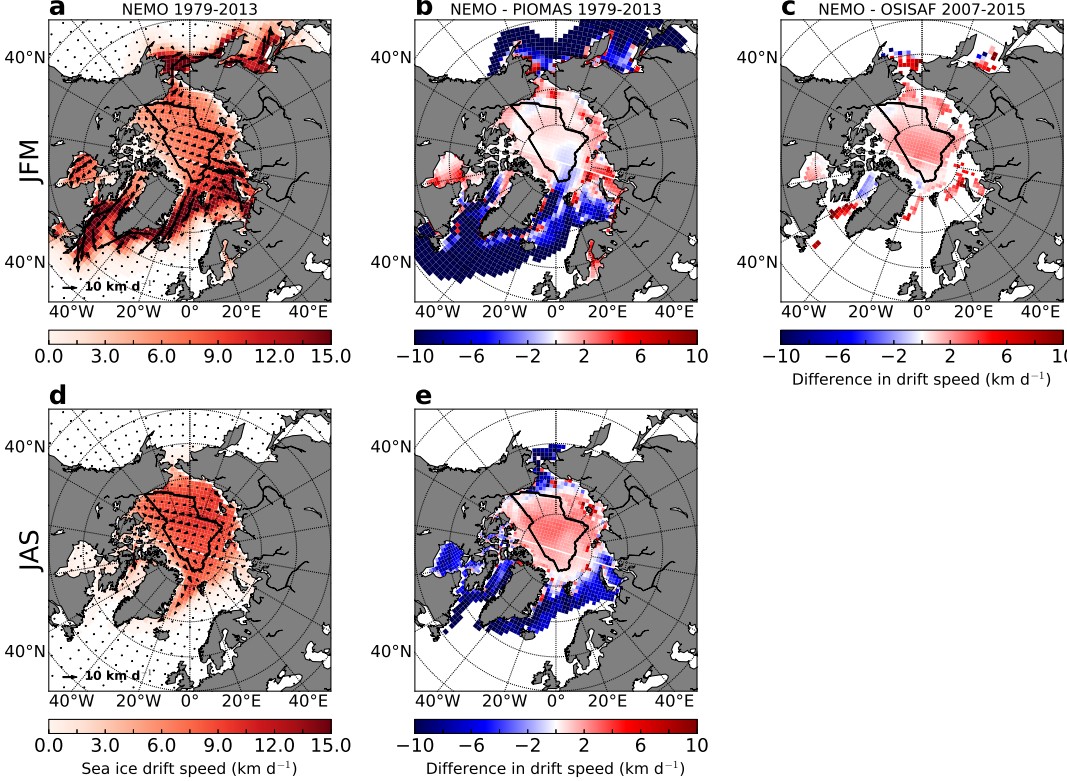

**Figure 6.** (a) Modelled (NEMO-LIM3.6) mean Arctic sea ice drift speed averaged over the winter months (JFM, i.e. January-February-March) of the period 1979-2013. (b) Difference in drift speed between NEMO-LIM3.6 and PIOMAS averaged over the winter months of the period 1979-2013. (c) Difference in drift speed between NEMO-LIM3.6 and OSI SAF averaged over the winter months of the period 2007-2015. (d), (e) Same as (a), (b) respectively for the summer months (JAS, i.e. July-August-September). OSI SAF drift data are not available in summer. The black polygon is the contour of the SCICEX box.



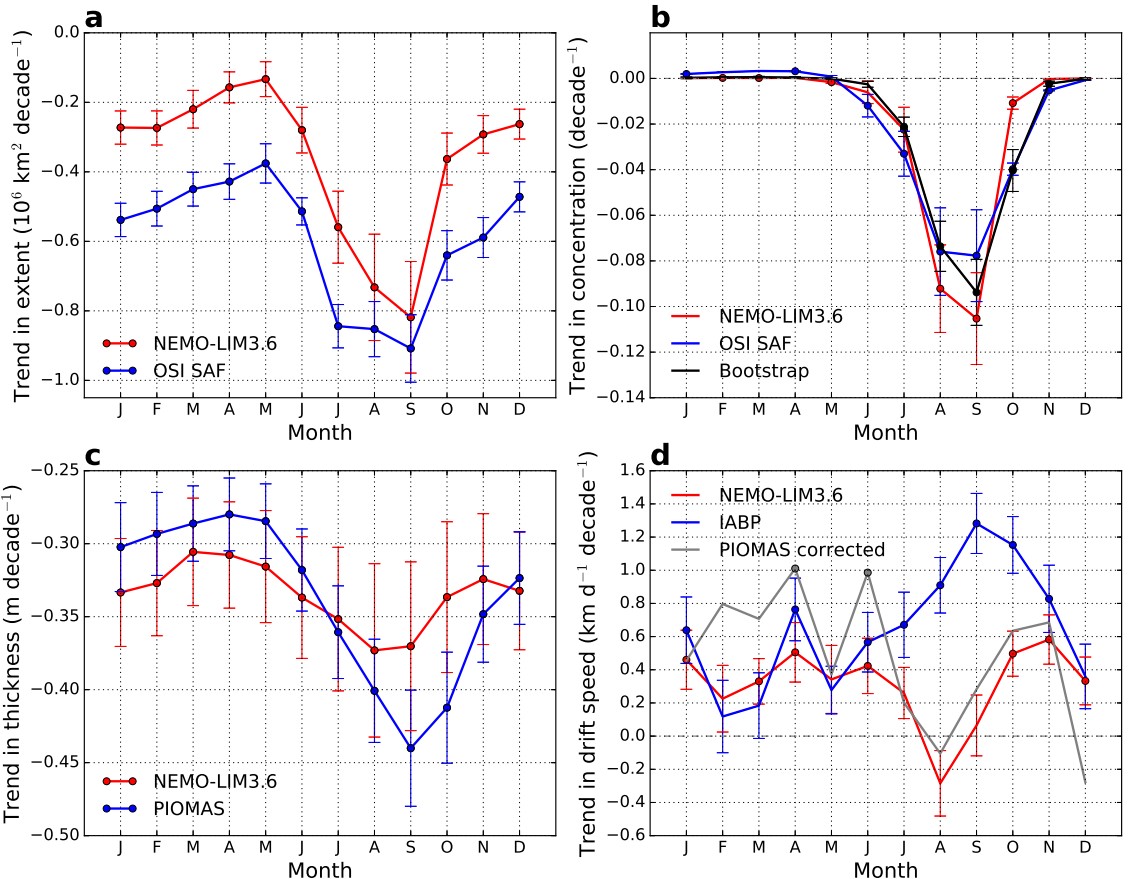

**Figure 7.** Modelled (NEMO-LIM3.6) and observed monthly mean seasonal cycles of trends in Arctic sea ice (a) extent, (b) concentration, (c) thickness and (d) drift speed averaged over the period 1979-2013. The spatial mean over the SCICEX box is represented in (b), (c) and (d). A dot is shown for monthly trends that are significant at the 5% level. Sources for observations: OSI SAF satellite data for extent and concentration, Bootstrap satellite data for concentration, PIOMAS reanalysis for thickness and drift speed, and IABP buoys for drift speed. PIOMAS drift speed data are computed from monthly velocity components and are multiplied by two to be comparable to other drift data. Error bars show the temporal standard deviation of monthly values.





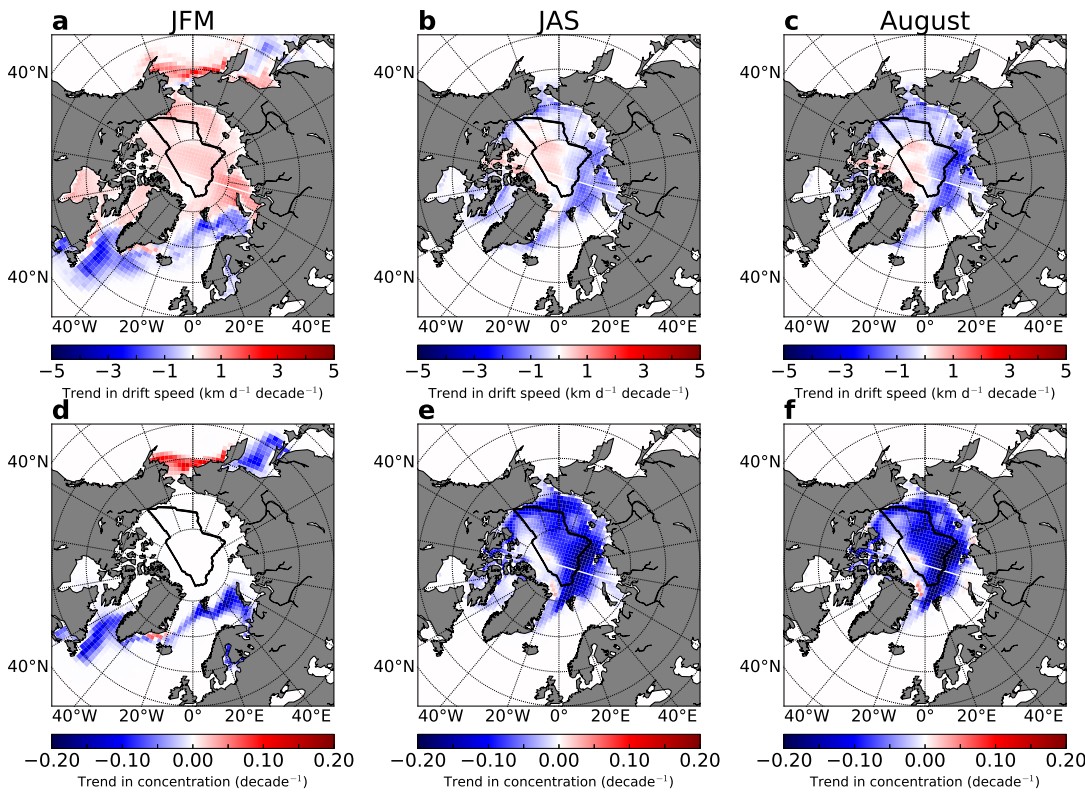

**Figure 8.** (a) Modelled (NEMO-LIM3.6) mean trend in Arctic sea ice drift speed averaged over the winter months (JFM, i.e. January-February-March) of the period 1979-2013. (b) Same as (a) for the summer months (JAS, i.e. July-August-September). (c) Same as (a) for August only. (d), (e), (f) Same as (a), (b), (c) respectively for sea ice concentration. The black polygon is the contour of the SCICEX box.

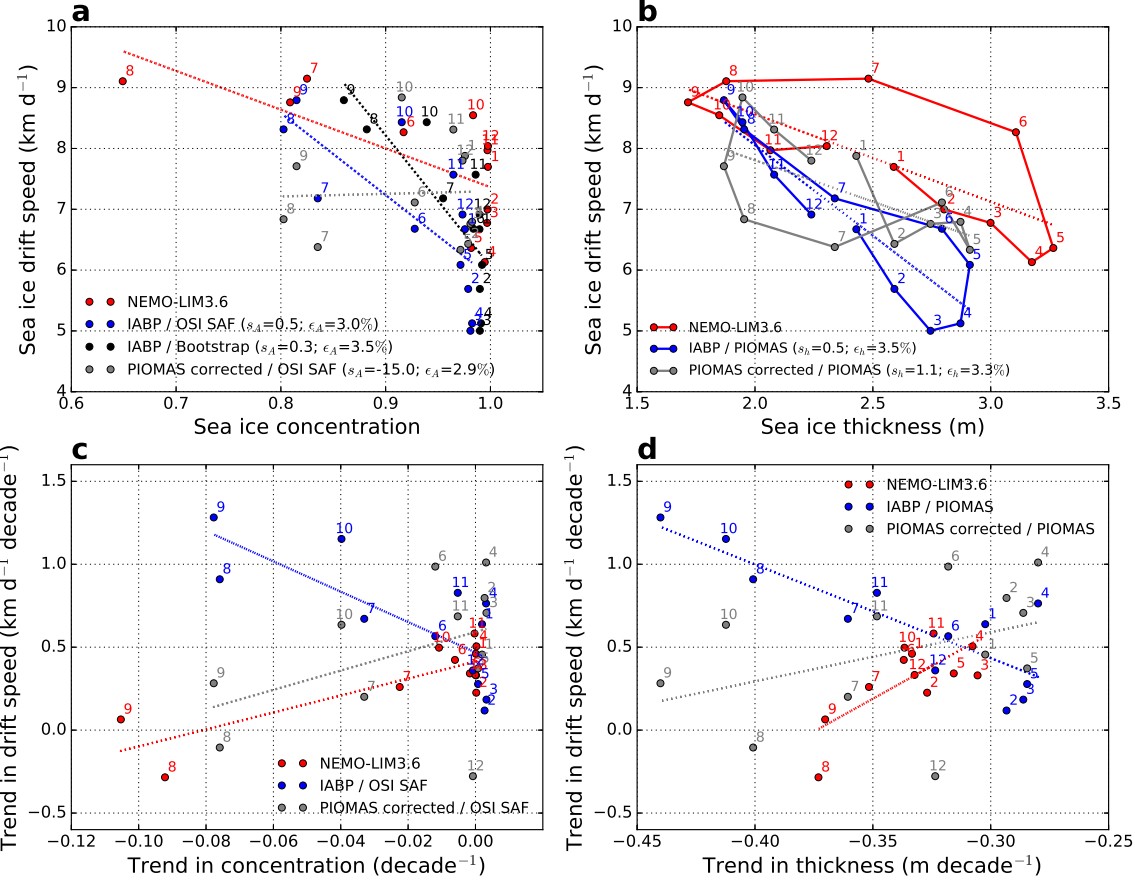

**Figure 9.** Scatter plots of modelled (NEMO-LIM3.6) and observed monthly mean sea ice drift speed against (a) concentration and (b) thickness temporally averaged over the period 1979-2013 and spatially averaged over the SCICEX box. (c) and (d) are similar plots for trends. Numbers denote months. Dotted lines show linear regressions. Sources for observations: IABP for drift speed, OSI SAF and Boostrap for concentration, PIOMAS for thickness and drift speed. Slope ratios and normalised distances between NEMO-LIM3.6 and the different observation datasets are shown in brackets in the legends of (a) and (b). PIOMAS drift speed data are computed from monthly velocity components and are multiplied by two to be comparable to other drift data.

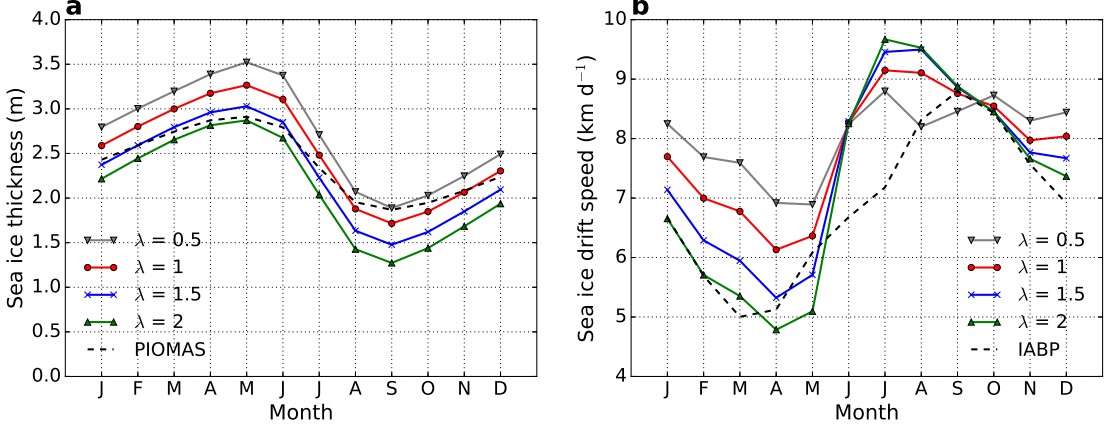

**Figure 10.** Modelled (NEMO-LIM3.6) monthly mean seasonal cycles of sea ice (a) thickness and (b) drift speed temporally averaged over the period 1979-2013 and spatially averaged over the SCICEX box for four different $\lambda$ values (see Eq. (1)). Observations are represented as dashed black lines (PIOMAS for thickness in (a) and IABP for drift speed in (b)).

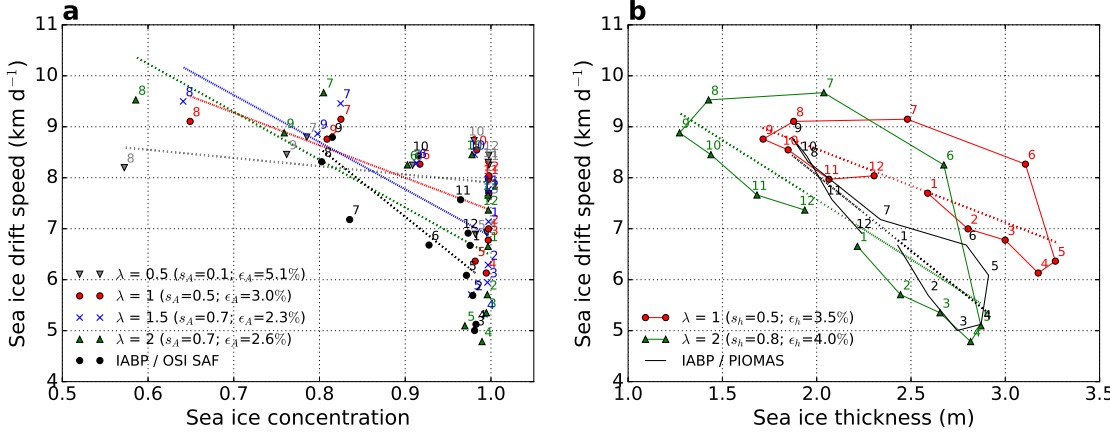

**Figure 11.** Scatter plots of modelled (NEMO-LIM3.6) monthly mean sea ice drift speed against sea ice (a) concentration and (b) thickness temporally averaged over the period 1979-2013 and spatially averaged over the SCICEX box for different $\lambda$ values (four values in (a) and two values in (b) for readability). Numbers denote months. Observations are represented in black. Dotted lines show linear regressions. Slope ratios and normalised distances between NEMO-LIM3.6 and the different observation datasets are shown in brackets in the legend.





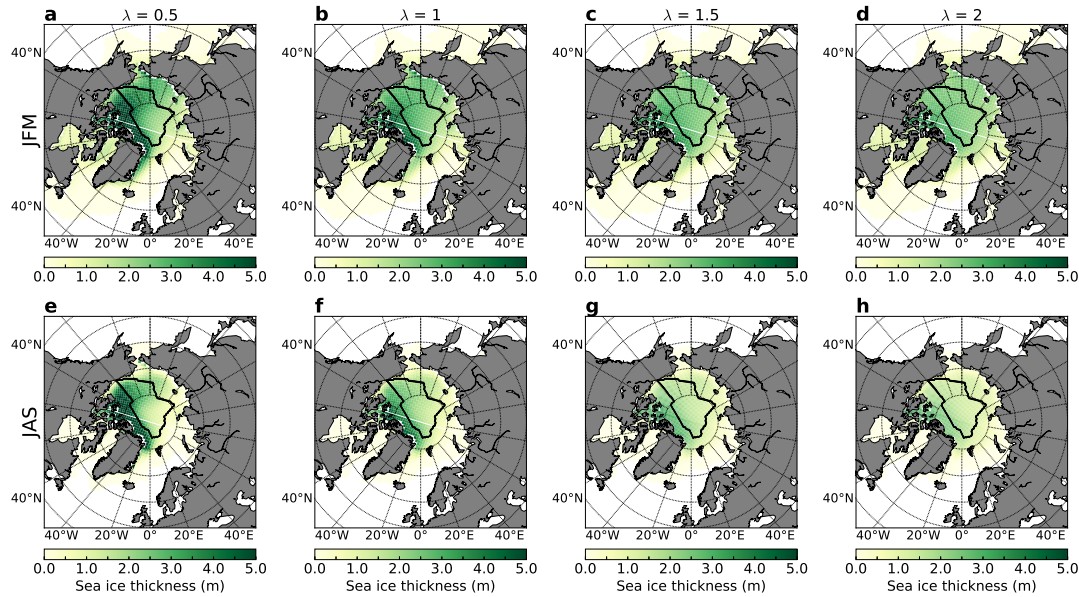

**Figure 12.** Modelled (NEMO-LIM3.6) mean Arctic sea ice thickness averaged over the winter months (JFM, i.e. January-February-March) of the period 1979-2013 for (a) $\lambda = 0.5$, (b) $\lambda = 1$, (c) $\lambda = 1.5$, (d) $\lambda = 2$. (e), (f), (g), (h) Same as (a), (b), (c), (d) respectively for the summer months (JAS, i.e. July-August-September). The black polygon is the contour of the SCICEX box.

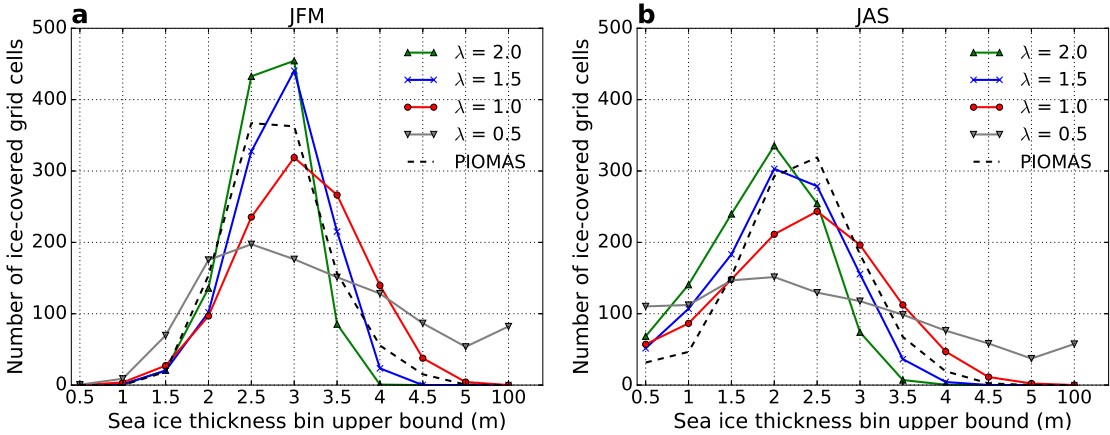

**Figure 13.** Number of ice-covered grid cells in each thickness bin temporally averaged over 1979-2013 and spatially averaged over the SCICEX box for both (a) winter (JFM, i.e. January-February-March) and (b) summer (JAS, i.e. July-August-September). Results are shown for NEMO-LIM3.6 (four different $\lambda$ values) and for PIOMAS reanalysis interpolated onto the ORCA1 grid. The x axis shows the upper bound of each thickness bin.



**Table 1.** Summary of published literature providing observational drift speed trend and its cause, as well as area and volume export of sea ice at Fram Strait.

| Reference | Drift speed | Cause of drift increase | Area export | Volume export |
|---|---|---|---|---|
| Häkkinen et al. (2008) | 1950-2006: significant positive trend | Wind | | |
| Spreen et al. (2009) | | | | 1990-2008: no significant change |
| Kwok (2009) | | | 1979-2007: no significant trend | |
| Rampal et al. (2009) | 1979-2007: +17% decade$^{-1}$ in winter +8.5% decade$^{-1}$ in summer | Ice strength | | |
| Spreen et al. (2011) | 1992-2009: +10.6% decade$^{-1}$ | Ice strength (first) Wind (second) | | |
| Gimbert et al. (2012) | | Ice strength | | |
| Polyakov et al. (2012) | | | 1979-1995: increase | |
| Vihma et al. (2012) | 1989-2009: increase | Ice strength (first) Wind (second) | | |
| Kwok et al. (2013) | 1982-2009: +6.2% decade$^{-1}$ in winter +3.6% decade$^{-1}$ in summer | Not wind | 1982-2009: small decrease | |
| Langehaug et al. (2013) | | | 1957-2005: small increase | |
| Döscher et al. (2014) | Increase | From 1990: ice strength Before 1990: wind | No long-term trend | Decrease |
| Olason and Notz (2014) | 1979-2011: +1.1 km d$^{-1}$ decade$^{-1}$ in summer +0.4 km d$^{-1}$ decade$^{-1}$ in winter | Ice strength | | |
| Krumpen et al. (2016) | | | 1980-2012: significant positive trend | |
| Smedsrud et al. (2016) | Increase | Wind | 1979-2014: +6% decade$^{-1}$ | |