# Peer review of "Interactions between Arctic sea ice drift and strength modelled by NEMO-LIM3.6"

_The Cryosphere, 2017_

## Referee Comment (RC1) · Anonymous Referee #1 · 23 May 2017

Overview and major comments:

In this paper, the authors analyse the results from the ice-ocean model NEMO-LIM3.6, forced with atmospheric reanalysis, in order to better understand the drift-strength feedback in the Arctic. Based on previous work the authors propose new metrics and use those, as well as other metrics and diagnostics to evaluate their model against observations and results from the PIOMAS model. They then discuss how their evaluation relates to the drift-strength feedback and do a sensitivity experiment to evaluate how ice strength in their model affects the modelled drift-strength feedback.

It's always nice to see modellers evaluate their model results against data and the authors should be commended for making the effort here. It was also nice to see an evaluation that goes beyond considering just the concentration and extent and I

enjoyed seeing that the authors are trying to push for new methods of analysing their model

My main reservation, though, regarding the paper is the premise of the drift-strength feedback, as presented here. In particular, the authors state that larger sea-ice drift leads to larger exports, but this does not seem to be the case. It is well established that the drift speed of ice in the Arctic is increasing, but at the same time there seems to be no clear increase in (Fram Strait) export. Some studies do find an increase, while others find no increase or a decline in the export. The authors themselves choose (very rightly I think) to cite Döscher et al., which say that there is no significant long-term trend in the area export and a slight decrease in the volume export (p. 2, l. 20 of the manuscript). Thus, we have established increase in the drift speed and no increase in export and we therefore cannot connect the "Drift" and "Export out of Arctic Basin" boxes in figure 1. This puts in question the premise of the paper and some of its contents (though not nearly all).

The reason we don't see an increase in export even if the drift speed increases is that the increase in drift speed is in the synoptic-scale back-and-forth movement of the ice, not the long-term, large-scale drift. This is highlighted by Olason and Notz when concentration is low, but it also seems to be the case when concentration is high.

I consider this a major shortcoming of the paper and recommend that the authors re-think and re-structure its contents. There is good material here which, with some re-structuring and extra work can be made into a good paper.

Minor comments:

p. 1 l. 11: You say "We demonstrate that . . . leading to lower heat conduction fluxes . . .", but there is no analysis of the fluxes provided. As it is you don't "demonstrate", but "suggest" or "speculate". An actual demonstration of this would be very interesting to see, especially since I don't think this is what's happening. I would think that higher ice strength results in less ridging which then results in less volume. This is the result of

Steele et al. (1997), as well as Flato and Hibler (1995) and I tend to think this is what you get as well.

p. 3 l. 29-32:

* You use P* = 20 kN/mˆ2. This is quite small. The "canonical" value of Hibler and Walsh is 27.3 and that was using daily forcing. What is the temporal resolution of your forcing? If it's something like every 6 hours then you should be using a larger value than Hibler and Walsh, not smaller. You need a reference for this value.

* You give no justification for the lambda parameter in equation (1). This is non-standard and requires at least a reference to back it up.

* Why don't you try different values of P* instead of changing lambda? It is well known to be an extremely uncertain parameter and I'm already suspicious of the value you use.

p. 4 l. 28: You should really calculate the model speed the same way the observation speed is calculated, not calculate a one-day average from a two-day observation. But the effect here is probably very small.

p. 4 l. 29: PIOMAS is a model, not observations, and I would like to ask you to please not treat it as observations. It has plenty of shortcomings and uncertainties all on its own.

p. 5 l. 20: I guess the paragraph on p. 4 l. 29 belongs here. Just keep in mind that even though Schweiger et al. (2011) is a very nice paper, then PIOMAS is not the truth. I would ask you to reduce considerably your reliance on PIOMAS in this study and try to compare to actual observations instead, as flawed as they may be. You also haven't considered the Rothrock et al. (2008) multiple regression model, which is well worth taking into account here.

p. 6 l. 6: This is not the right reasoning for choosing daily time scales. With daily time scales you capture synoptic-scale variability, but with monthly time scales you average

these out and capture the longer-term, large-scale drift.

p. 6 l. 8: Given my comment above it should be clear that you cannot use the monthly values from PIOMAS in this way. They contain different physics and you can't just scale with factor two!

p. 6 l. 21: From here on out this section becomes increasingly hard to understand. I had to re-read and then re-read again to completely understand which metrics and diagnostics you use. It's all there, but you're making your reader work way too hard to get the point. Please rewrite and try to make it clearer and better organised.

p. 6 l. 28: The novelty here is really that you use this method as a way to evaluate your model.

p. 6 l. 29: You don't normalise with wind friction speed, but Olason and Notz (2014) say they do this to take atmospheric stability into account. It is interesting that you find that this is not necessary, it is not what they find. However, I don't understand why you don't normalise with the 10 m wind speed at least, since we know there should be a close correlation between drift speed and wind speed. My main concern, however, is that your figure 9b gives a completely different shape for the curve than figure 6 from Olason and Notz (2014). Why is that?

p. 7 l. 9: These are probably good metrics you've developed, but you don't use them enough and you don't discuss them enough to make me want to use them too.

p. 7 l. 15: Mention (again) the period you average over.

p. 7 l. 16: What are the (main) differences between you set up and Rousset's et al. (2015)? If it's just the resolution then remind the reader which resolution you use.

p. 7 l. 27 (all paragraph): I'm concerned that you rely too much on comparison with PIOMAS. Again, it's only a model so you should try hard(er) to compare to observations before resorting to comparing with PIOMAS.

[Figure]

p. 8 l. 6 (the paragraph and this section in general) You jump a lot between the SCICEX box and your "wider domain" and I'm having trouble keeping up. Try to decide which is more important, stick to it and mention the other one only when necessary.

p.8 l. 26: What conclusion should I draw from this paragraph? Is the trend significant or a post-processing glitch?

p. 10 l. 6: You need a justification for using lambda and not tuning P*

p. 10 l. 14: It's not counter-intuitive to me, as I mentioned earlier when commenting on the abstract.

p. 10 l. 25: It's not really a hysteresis loop. Physically the drift speed depends on ice thickness only when the concentration is high, so the change in drift speed only relates to the change in thickness in winter.

p. 11 l. 2: Your heat-flux theory contradicts the results of Steel et al. and Flato and Hibler. You need to show that it's true by actually showing the ocean-atmosphere heat flux and analysing that.

p. 11 l. 22: It's only a physical correlation in winter. See my comment for p. 10 l. 25

p. 11 l. 28: I don't know what you mean by "large-scale effects"

p. 13 l. 16: I can draw no concrete conclusions from this sub-section

Figures 5, 6, 8, and 12: These figures are very small and hard to read. It would be better if they focused on the central Arctic. I don't think you would lose much information doing that. Figure 6 is particularly hard to read and I can't see the directions of the vectors at all. It would also be nicer to have the magnitude and direction of the difference, rather than the way it's done now.

---

## Referee Comment (RC2) · Anonymous Referee #2 · 2 Jun 2017

Comments on "Arctic sea ice drift-strength feedback modelled by NEMO-LIM3.6" by Docquier et al.

**1. Summary**

The authors conducted ocean-sea ice model NEMO-LIM3.6 numerical experiments to investigate interaction and/or feedback mechanisms between sea ice drift speed and ice strength, focusing on the Arctic Ocean. As measures of ice strength, the authors employ ice concentration and thickness, and then examined relation between ice drift speed and them. In order to assess the model performance, sea ice observation data derived from satellite and ice-tethered buoys (ice concentration, ice drift speed and ice thickness) were exploited. The authors introduced a systematic model validation method based on sea ice diagnostics (ice extent, ice concentration, ice thickness and ice drift speed) as well as process-based diagnostics (relation between different ice properties, e.g., ice drift speed and concentration). They also introduced metrics which quantify modeled or observed ice dynamics as scalar variables. Using these diagnostic and metric approaches, the author assessed model performance, and then conducted sensitivity experiments varying ice strength parameter to see the effect of ice strength change on the relation between drift speed and ice concentration (or ice thickness) simulated in the model.

**2. General comments**

The authors provided a good review regarding relation between ice drift speed and ice concentration (and thickness) in the Arctic Ocean, mainly focusing on observational studies. The diagnostics and metrics used to validate model performance and the thorough assessments on the simulated ice properties are very helpful to assess the capability and limitation of the model, although I have an additional suggestion regarding ice thickness evaluation (see comments below). The results of the sensitivity experiment, in which ice strength parameterization is changed, are interesting to me (and probably informative to many sea ice modellers), while the whole strategy used to assess the feedback between ice drift and other ice properties seems to me not always suitable nor convincing for the purpose of the study. As far as I know, "feedback" is a term originally introduced to describe an electric circuit which has a recursive input, and is widely used in other study area to explain similar concept. Also in geo-scientific studies, the term "feedback" is widely used to explain a recursive interaction between different processes or subsequent chain of phenomena, sometimes without clear definition nor quantification. Although the authors invoked the term "feedback", the strategy does not seems to be designed so as to extract a feedback mechanism from a complicated system. For this reason, I would recommend to change the focus of the study to more specific issues (e.g., sensitivity of ice strength parameterization to ice drift, concentration and thickness) or to reorganize experiment design so as to quantify the feedback between them. Even without feedback issue, the manuscript contains interesting model results.

**3. Major point**

- The strategy used to extract (and quantify) the drift-strength feedback is not always suitable nor convincing. I think more consideration is necessary about how to quantify a "feedback". Since ice drift, concentration and thickness are described by a set of simultaneous partial differential equations in numerical models, it is a matter of course that there is some sort of 'feedback' between them. An important point is how to extract and quantify a feedback in a simple formula so as to abstract the essence of the complicated system. Mathematically, a feedback between two variables (most simple case) can be described by a set of equations,

$$\frac{\mathrm{d}\overline{|V|}}{\mathrm{dt}} = F(\overline{T}) \tag{1}$$

$$\frac{\mathrm{d}\overline{T}}{\mathrm{dt}} = G(\overline{|V|}) \tag{2}$$

where $\overline{|V|}$ and $\overline{T}$ are respectively mean ice drift velocity and ice strength (or thickness) in the present case (e.g., $\overline{|V|} = \frac{1}{TS}\iint |V| dxdydt$, $T$: one month or one year period, $S$: SCICEX box). $F$ and $G$ are the functions describing a feedback. The equations mean that the temporal evolution of mean ice drift is controlled (or affected) by mean ice strength, while at the same time, the evolution of ice strength is also controlled by drift speed. The most simple solution is an exponential formula, $Ce^{\alpha t}$, and then temporal evolution of $\overline{|V|}$ and $\overline{T}$ depends on $\alpha$. The system has a positive feedback if $\alpha > 0$, while a negative feedback if $\alpha < 0$. If the authors intend to clarify and quantify the feedback mechanism between ice drift and thickness, a definite formula (not necessarily means complicated formula) in such a simple theoretical framework should be provided. Otherwise, we cannot learn anything about the quantitative features of the feedback between ice drift and thickness (or concentration) at all. Note that the system also needs a higher order stabilizing term to prevent exponential growth of a positive feedback. (At least discussion for such a damping mechanism is necessary)

- The design of the sensitivity experiment seems to me not suitable for examining the effect of ice strength change on ice drift, ice thickness and concentration. Since the authors change the ice strength, $P$, by changing exponent of ice thickness $h$, the associated change of ice strength has an opposite sign between $h > 1$ and $h < 1$. This is easily confirmed by calculating $P$ value for different $\lambda$ used in the sensitivity experiment: if $h > 1$, $P_{\lambda=2}$ is larger than $P_{\lambda=1}$, while $P_{\lambda=2}$ is smaller than $P_{\lambda=1}$ if $h < 1$ (this is also confirmed by closely looking Fig. 2). Due to this experiment design, we cannot directly relate the change of parameter $\lambda$ to the change of ice strength and therefore, to changes of the ice thickness and ice concentration, which makes interpretation of the results complicated and difficult (There are many mis-interpretation in sec. 3.3 due to this fact, see the specific points below). Generally speaking, changing exponent of $h$ in the ice strength equation is not equivalent to changing ice strength, but to changing sensitivity of ice strength to $h$. My recommendation is to use more simple formula for the sensitivity experiment (e.g., $P = P^* \lambda h \exp[-C (1 - A)]$, which is equivalent to change $P^*$), or to redo analyses based on the actual ice strength $P$ used in the model (i.e., calculate $\overline{P} = \frac{1}{TS}\iint P(x,y,t) dxdydt$ and examine relation between $\overline{P}$ and modeled ice properties; ice drift, ice concentration and thickness).

- I admire the systematic approach used in this study to assess the model performance using observational data, whereas I have a concern about the use of ice drift and thickness data derived from PIOMAS, instead of direct observations. The authors did not utilize two important dataset for ice drift and thickness, which have sufficient spatial and temporal coverage for the present study. One is ice drift data provided from Colorado University group (Tschudi et al., 2016), the other is the long-term ice thickness estimate by Lindsay and Schweiger (2016). Both estimates provide error of the estimates as well. Since these data were derived from in-situ and satellite measurements while PIOMAS did not assimilate ice drift and thickness, I recommend to use these two data instead of PIOMAS simulation. Note that the bias (or error) of ice drift field in Tschudi et al. (2016) reported by Szanyi et al. (2016) is a crucial issue, only if the data is used for divergence/convergence calculation. Since the divergence/convergence features reported by Szanyi et al. (2016) always appear as a divergence/convergence pair around buoy-data merged location, a spatial averaging (e.g., SCICEX box) can eliminate the error.

4. Specific points / additional comments

- Page 2, line 1-8: I would suggest the author to use the term 'feedback' more carefully, particularly when mentioning 'positive feedback'. Since a positive feedback has an exponential growth feature by its definition, it always needs damping or stabilizing mechanism in higher order, when the concept is used to explain things occurring in the nature.

- Page 2, line 9-22: I appreciate the good summary of the former studies here, which is useful to survey the current status of our understanding on this issue. On the other hand, I am a little bit skeptical to provide conceptual illustration like Fig. 1, since such an illustration sometimes goes out of authors' control if once published, even if each arrow in the figure is not really examined.

- Page 3, line 29-30: Why the authors conducted the sensitivity experiment by changing exponent of $h$ in the ice strength equation, instead of changing $P^*$ or $C$? If the authors intend to increase ice strength in the entire $h$ range, changing $P^*$ seems to be more suitable approach (as many modellers do). I request more explanation.

- Page 4, line 1-7: Since we cannot directly relate the increase of $\lambda$ to increase of ice strength in the present experiment design, more careful explanation is needed. For example, ice thickness during the summer season in the SCICEX box may thinner than 1 m, at least part of the area. In this case, an increase of $\lambda$ leads to decrease of ice strength.

- Page 4, line 26-34: Why the authors did apply ice drift data provided from PIOMAS instead of satellite- and buoy-based data as a long-term ice drift observation? Polar Pathfinder Daily 25 km EASE-Grid Sea Ice Motion Vectors (Tschudi et al., 2016, now version 3 is available via NSIDC website) provides Arctic-wide long time series from 1979 to present. Although PIOMAS reasonably reproduced sea ice extent and ice concentration field as a result of ice concentration assimilation, there is no reason to believe that ice drift and thickness from PIOMAS are consistent with observation, since these variables are not assimilated. I think the ice drift data from PIOMAS should be dealt with a great care (Since PIOMAS applies strong constraint on ice concentration by an optimal interpolation, the simulated ice field never breaks down, even if ice drift is totally unrealistic). Use of Polar Pathfinder data is much more reliable approach, since the uncertainty of the estimates are also provided. The problem of the Polar Pathfinder data reported by Szanyi et al. (2016) is not a critical issue for the present application (see my major point as well).

- Page 5, line 20-24: For long-term ice thickness estimates in the Arctic Ocean, I strongly recommend to use the estimate presented in Lindsay and Schweiger [2015]. They provided an empirical function describing the spatially and temporally varying ice thickness field over the Arctic Ocean, by exploiting all available in-situ and satellite measurements of ice thickness. As far as I know this is the most reliable long-term estimate of ice thickness field based on measurement for the time being (One can calculate seasonally varying long-term ice thickness field by the description in Lindsay and Schweiger).

- Page 6, line 1-4: How did the authors define the daily ice drift in Eq. (2)? $u_d = \dfrac{1}{T_{day}} \displaystyle\int_{t_1}^{t_2} v(t)\,dt$

(i.e., temporal average of instant velocity in x and y direction) or $u_d = \dfrac{\left| x_{t_2} - x_{t_1} \right|}{T_{day}}$ (zonal or

meridional displacement for 24 hours)? The definition of satellite- or buoy-derived ice drift is the

latter. I don't think the difference between the two definition is large, but it would be nice if the authors clarify their definition for comparison with other model results.

- Page 7, line 27-30, Page 8, line 1-5: I think the comparison with PIOMAS thickness data provides useful insights, while I am skeptical to regard PIOMAS thickness data as substitution for observation, since we cannot distinguish bias or error of the estimates, which are always provided for observational data.

- Page 8, line 1: Do the authors have an explanation why the peak shifts?

- Page 8, line 23-24: How the IABP ice drift data were processed to calculate spatial (and temporal) average over the SCICEX box? Since the spatial coverage of the buoy data is not sufficient, one needs to interpolate/extrapolate the data. How did the author define influential radius of the buoy data? How much uncertainty should we expect from the interpolation/extrapolation process? Since the IABP data averaged over the SCICEX box is an important measure to validate the model, a description is necessary.

- Page 9, line 2-3: Why the authors show the relationships in terms of mean seasonal cycle, not by scatter plots based on the relations in each month? I mean, for example, a scatter plot for drift-concentration relation in each month can more clearly show the validity of the regression line.

Page 9, line 16-17: Why the authors did not apply normalization by wind stress? Since the wind stress may differ between each month, it is difficult to derive a general relation between drift speed and other variables. If there is a reasonable explanation for not to apply normalization, please describe in the text.

Page 9, line 30-31: I think the use of 'feedback' in this sentence is not appropriate. The result in this section (sec. 3.2) shows that there is a (linear) relationship between drift speed and strength (thickness or concentration) as equilibrium states (A feedback system does not reach an equilibrium state unless $\alpha = 0$, or having oscillating solution. see my major point). I don't mean the analysis in this section is meaningless, but is not appropriate to show the existence of feedback (see also major point).

Page 10, line 9-20: There are a number of incorrect sentences here. Please keep in mind that higher $\lambda$ does not directly correspond to larger ice strength, due to the current formulation, Eq. (1). Particularly, it is not true, when discussing relation between ice drift and thickness (or concentration) in summer season. The thickness may thinner than 1 m at least part of the SCICEX box.

Page 10, line 16-20: I think this interpretation is wrong, probably due to the fact which I described in major point. Since the ice thickness during summer season is close to (or even smaller than 1 m), the ice strength for larger $\lambda$ becomes smaller than that for smaller $\lambda$. Therefore, the larger drift speed for larger $\lambda$ can be simply the result of smaller ice strength.

Page 10, line 32-33: I would say the result is not counter-intuitive. Note that increase of $\lambda$ leads to smaller ice strength in $h < 1$ area, it means ice can be easily deformed or compressed in $h < 1$ area, compared to small $\lambda$.

Page 10, line 4 - page 11, line 8: Section 3.3 needs additional figure showing the relation between $\lambda$ and mean ice strength in SCICEX box (a seasonal cycle for each $\lambda$ should be shown), otherwise, we cannot relate ice strength with ice thickness (and concentration) nor examine the relation between ice strength and ice drift.

Page 10, line 33 - page 11 line 6: I think the reason for the increase of modal thickness for larger λ (Fig. 13) can be simply explained. The experiment with increased λ has larger ice strength for h > 1 (winter season), which prevents ice thickening due to ridging, while it has smaller ice strength for h < 1 (summer season), which enhance ice thickening due to ridging. As a result, larger λ leads to larger peak at modal thickness.

Page 11, line 14-15: I would say that this study also did not 'quantify' the magnitude of the drift-strength feedback. To quantify a feedback, one should present growth rate of the feedback.

Page 11, line 23-24: Due to the analyses without normalization by wind stress, it is difficult to distinguish the reason for the hysteresis loop shown in Fig. 9b and Fig. 11b. Can the hysteresis loop be observed if the ice drift speed is normalized by wind stress?

Page 12, line 14 - page 13, line 15: As pointed in major point, I think the basic strategy for the sensitivity experiment and analyses are not suitable for quantifying 'feedback'. If the authors intend to quantify 'feedback', the entire framework should be reconsidered..

Page 14, line 16-18: Why such hysteresis loops are observed in the modeled sea ice? Since observation (IABP) does not show such a feature, I guess this is an artifact coming from insufficient modeled physics..

Page 14, line 21-28: The summary provided here should be reconsidered, by taking the second paragraph of the major point into account. Although the authors generally provided descriptive result of their analyses on ice drift - thickness relations, they did not show any results on thermodynamic analyses. Therefore I cannot follow nor rely on the arguments associated with thermodynamic effects..

**5. Reference**

Lindsay, R. and A. Schweiger, 2015: Arctic sea ice thickness loss determined using subsurface, aircraft, and satellite observations, The Cryosphere, 9, 269-283, doi:10.5194/tc-9-269-2015.

Szanyi, S., J. V. Lukovich, D. G. Barber, G. Haller, 2016: Persistent artifacts in the NSIDC ice motion data set and their implications for analysis, Geophys. Res. Lett., pp. 10800-10807, doi:10.1002/2016GL069799.

Tschudi, M., C. Fowler, J. Maslanik, J. S. Stewart, and W. Meier, 2016: Polar Pathfinder Daily 25 km EASE-Grid Sea Ice Motion Vectors. Version 3. Boulder, Colorado, USA. (https://nsidc.org/data/docs/daac/nsidc0116_icemotion.gd.html)

---

## Author Comment (AC1) · 13 Jul 2017

**Reply to Referee #1**

**Interactions between Arctic sea ice drift and strength modelled by NEMO-LIM3.6**

**Docquier *et al.* (2017), tc-2017-60**

We would like to thank Referee #1 for his/her very constructive feedback, which has helped us improve the paper quality. Below we present our detailed responses to the comments and suggestions proposed by the reviewer in **blue**. The corresponding corrections are in **blue** in the revised manuscript.

**1. Overview and major comments**

In this paper, the authors analyse the results from the ice-ocean model NEMO-LIM3.6, forced with atmospheric reanalysis, in order to better understand the drift-strength feedback in the Arctic. Based on previous work the authors propose new metrics and use those, as well as other metrics and diagnostics to evaluate their model against observations and results from the PIOMAS model. They then discuss how their evaluation relates to the drift-strength feedback and do a sensitivity experiment to evaluate how ice strength in their model affects the modelled drift-strength feedback.

It's always nice to see modellers evaluate their model results against data and the authors should be commended for making the effort here. It was also nice to see an evaluation that goes beyond considering just the concentration and extent and I enjoyed seeing that the authors are trying to push for new methods of analysing their model.

My main reservation, though, regarding the paper is the premise of the drift-strength feedback, as presented here. In particular, the authors state that larger sea-ice drift leads to larger exports, but this does not seem to be the case. It is well established that the drift speed of ice in the Arctic is increasing, but at the same time there seems to be no clear increase in (Fram Strait) export. Some studies do find an increase, while others find no increase or a decline in the export. The authors themselves choose (very rightly I think) to cite Döscher et al., which say that there is no significant long-term trend in the area export and a slight decrease in the volume export (p. 2, l. 20 of the manuscript). Thus, we have established increase in the drift speed and no increase in export and we therefore cannot connect the "Drift" and "Export out of Arctic Basin" boxes in figure 1. This puts in question the premise of the paper and some of its contents (though not nearly all).

The reason we don't see an increase in export even if the drift speed increases is that the increase in drift speed is in the synoptic-scale back-and-forth movement of the ice, not the long-term, large-scale drift. This is highlighted by Olason and Notz when concentration is low, but it also seems to be the case when concentration is high. I consider this a major shortcoming of the paper and recommend that the authors re-think and re-structure its contents. There is good material here which, with some re-structuring and extra work can be made into a good paper.

We agree with the reviewer that the drift-strength feedback that we present in our study has not been formally demonstrated by any previous study. Rampal et al. (2011) suggest it might be an important feedback but its existence has never been proved formally. Due to the lack of observational evidence to confirm this feedback, the results we obtain with our sensitivity experiments and the remarks from both reviewers, we decided to change the focus of our article. We now concentrate more on the interactions between sea ice drift speed and strength (concentration and thickness) rather than on the feedback itself. Please see also our response to the first major point of Referee #2.

**2. Minor comments**

- p. 1 l. 11: You say "We demonstrate that ... leading to lower heat conduction fluxes ...", but there is no analysis of the fluxes provided. As it is you don't "demonstrate", but "suggest" or "speculate". An actual demonstration of this would be very interesting to see, especially since I don't think this is what's happening. I would think that higher ice strength results in less ridging which then results in less volume. This is the result of Steele et al. (1997), as well as Flato and Hibler (1995) and I tend to think this is what you get as well.

We do not have the model outputs of heat conduction flux for the sensitivity experiments with varying $\lambda$. However, we performed new sensitivity experiments in which we vary P* based on the comments from both reviewers, and for these experiments we made sure to compute heat conduction fluxes. The main results are:

- lower P* leads to higher average ice thickness (Fig. 9a in the revised manuscript) and higher ice thickness heterogeneity in space (Fig. 12 in the revised manuscript), which is in agreement with our previous $\lambda$ experiments
- lower P* leads to lower heat conduction fluxes and lower thermodynamic ice production (Fig. R1 below), which is not in agreement with our initial hypothesis.

[Figure]

*Fig. R1: Modelled (NEMO-LIM3.6) monthly mean seasonal cycles of (a) heat conduction flux at the ice bottom (positive from the ocean to the atmosphere) and (b) thermodynamic ice production temporally averaged over the period 1979-2013 and spatially averaged over the SCICEX box for five different P* values.*

Therefore, our initial hypothesis (lower initial ice strength leads to higher sea ice thickness heterogeneity, which results in higher heat conduction flux and higher ice production, hence larger ice thickness) is not the right reasoning that explains why ice thickness is higher with lower initial ice strength. We agree with the reviewer that the process is simpler: lower ice strength leads to higher deformation and more ice piling up (Fig. 11 and Table 2 in the revised manuscript), which results in higher ice thickness. We adapted the manuscript accordingly.

It is important to note that the results obtained by Steele et al. (1997) and Flato and Hibler (1995) arise from a different experimental setup than the one used in our study and do not exactly support the hypothesis that higher ice strength leads to less ridging and less volume, as we discuss now.

In the former study (Steele et al., 1997), the standard value $P^* = 27.5$ kN/m² is used in a sea ice model based on Hibler (1979) with a grid resolution of 40 km. This standard value is decreased and increased by a factor of 5 respectively. In the case of $P^*$ decreased by a factor of 5, the mean ice motion is faster (the ice is nearly in free drift) and the mean wintertime ice thickness is 35% higher than in the standard case. Increasing $P^*$ by a factor of 5 locks sea ice and motion of sea ice ceases, which produces an ice thickness that is essentially determined by equilibrium thermodynamics. In this case, the mean wintertime ice thickness is also slightly higher than in the standard case (by about 6%). A range of further sensitivity experiments performed by Steele et al. (1997) shows that the dependence of mean ice thickness h on $P^*$ is nonlinear (see their Fig. 16), with a sharp decrease of h with increasing $P^*$ for $P^* <= 27.5$ kN/m² and then a slight increase of h with increasing $P^*$ for $P^* > 55$ kN/m². Furthermore, ridging is not mentioned in their study. Therefore, not only the experimental setup of Steele et al. (1997) is different from ours, but also the results show a nonlinearity that we do not find (maybe because our λ values and $P^*$ values were not increased enough). The comparison of these results with our results is now discussed in our revised manuscript (Section 4.1).

In the second study, Flato and Hibler (1995) use a sea ice model based on the thickness distribution theory of Thorndike et al. (1975), which has a resolution of 160 km (which is much lower than the one used in our study), and perform sensitivity experiments by varying different ridging parameters. Since these experiments use a different model and a different experimental setup, we think it is difficult to perform a detailed comparison of their results with ours. But the results of Flato and Hibler (1995) do generally support the idea that more sea ice ridging leads to greater sea ice thickness.

- p. 3 l. 29-32:

* You use $P^* = 20$ kN/m^2. This is quite small. The "canonical" value of Hibler and Walsh is 27.3 and that was using daily forcing. What is the temporal resolution of your forcing? If it's something like every 6 hours then you should be using a larger value than Hibler and Walsh, not smaller. You need a reference for this value.

We agree with the reviewer that the value of $P^* = 20$ kN/m² used here is smaller than the value of $P^* = 27.5$ kN/m² found in Hibler and Walsh (1982), which provides the best agreement between their model and observations in terms of mean drift rates. However, $P^* = 20$ kN/m² is the commonly used value in the NEMO-LIM model and has been chosen via a tuning of mean sea ice

thickness and mean Fram strait ice export (Vancoppenolle, personal communication). It is also the value used in the viscous-plastic models of the Sea Ice Model Intercomparison Project (SIMIP) (Kreyscher et al., 1997) as well as in other modelling studies (Lipscomb et al., 2007; Juricke et al., 2013). Tremblay and Hakakian (2006) find that the most likely value of P* lies in the range 30-45 kN/m² based on satellite observations. Therefore, a wide range of P* values is used in the literature and not a single value is considered as a reference (Feltham, 2008). Since our new sensitivity experiments consider different values of P*, the precise choice of P* is no longer a concern. This information has been added to the revised manuscript (Section 2.1). The temporal resolution of our forcing (DFS5.2) is 6 hours.

* You give no justification for the lambda parameter in equation (1). This is non-standard and requires at least a reference to back it up.

Based on the comments from both reviewers, we performed new sensitivity experiments in which P* is varied. We decided not to show the results of the $\lambda$ experiments anymore. We only discuss them in Section 4.2.

The main goal of introducing a $\lambda$ parameter was to test the impact of a change in the strength parameterisation on sea ice drift speed and thickness. Previous studies have performed similar tests by using a square dependence of ice strength on thickness (Overland and Pease, 1988; Häkkinen and Mellor, 1992). Moreover, in the CICE model the ice strength P increases as a proportion of $h^{1.5}$ instead of h in order to improve the physical realism (Lipscomb et al., 2007). According to Leppäranta (2011), the value of $\lambda$ is an open question (just as P*). Other reasons for choosing $\lambda$ experiments are also given in response to the second major point of Referee #2.

* Why don't you try different values of P* instead of changing lambda? It is well known to be an extremely uncertain parameter and I'm already suspicious of the value you use.

As said earlier, we performed new sensitivity experiments in which we vary P* based on values found in the literature. We describe these experiments in the revised manuscript (Section 2.1). The chosen values provide an ice strength P range that is comparable to the $\lambda$ experiments:

- P* = 5.5 kN/m²: lowest value used by Steele et al. (1997) in their model sensitivity study, corresponding to 27.5 kN/m² divided by 5; this experiment is comparable to $\lambda$ = 0.5 in terms of strength-thickness dependence
- P* = 20 kN/m²: value commonly used in NEMO-LIM3.6 and other modelling studies; experiment comparable to $\lambda$ = 1
- P* = 27.5 kN/m²: reference value found by Hibler and Walsh (1982)
- P* = 45 kN/m²: highest value of the likely range found by Tremblay and Hakakian (2006) based on satellite sea ice drift observations; experiment comparable to $\lambda$ = 1.5
- P* = 100 kN/m²: value providing ice strength comparable to $\lambda$ = 2 and close to the highest value of Steele et al. (1997), i.e. 27.5 x 5 = 137.5 kN/m².

The main result of these experiments is similar to our $\lambda$ experiments, i.e. ice thickness increases with decreasing P* (Fig. 9a in the revised manuscript, to compare to Fig. 10a in the previous version of the manuscript).

- p. 4 l. 28: You should really calculate the model speed the same way the observation speed is calculated, not calculate a one-day average from a two-day observation. But the effect here is probably very small.

**We also computed the modelled sea ice drift speed the same way OSI SAF observations provide drift speed (i.e. two-day average) but did not find any significant difference compared to our method. Moreover, since we mainly compare our modelled drift speed to IABP buoy data, for which the temporal coverage and resolution are higher than OSI SAF, we prefer keeping the daily temporal resolution. This information has been added in the revised manuscript.**

- p. 4 l. 29: PIOMAS is a model, not observations, and I would like to ask you to please not treat it as observations. It has plenty of shortcomings and uncertainties all on its own.

**We agree with the reviewer that PIOMAS is not observations and we do not aim to treat it as such. The title of Section 2.2 ('Observations') is probably misleading since we include a brief description of PIOMAS into this section: the goal is more to put all data against which we evaluate our model in the same section rather than providing observations strictly speaking. We renamed Section 2.2 as 'Reference products'.**

**We also agree that PIOMAS has shortcomings and uncertainties. We now also use ULS submarine observations for sea ice thickness (the multiple regression model of Rothrock et al. [2008]) as well as the merged product of Tschudi et al. (2016) for sea ice drift speed. The manuscript has been revised accordingly.**

**However, observations also have uncertainties and suffer from sparse temporal and spatial coverage, especially for sea ice thickness (Stroeve et al., 2014; Zygmuntowska et al., 2014; Lindsay and Schweiger, 2015). Upward-looking sonar (ULS) measurements cover the period 1979-2005 but have incomplete spatial coverage and limited records for each year. Airborne (e.g. IceBridge) and satellite (e.g. ICESat, CryoSat) measurements only cover the recent period with very short temporal coverage for ICESat and limited spatial coverage for IceBridge. Lindsay and Schweiger (2015) conclude that 'more research to understand, characterize, and correct these errors [in sea ice thickness measurements] is clearly required before we can homogenize the observational ice thickness record'.**

**Furthermore, Schweiger et al. (2011) find that PIOMAS ice thickness estimates agree well with ICESat observations in the area for which submarine data are available, i.e. the SCICEX box. Therefore, we think PIOMAS still represents a valuable tool against which we can compare our modelled sea ice thickness since we use the SCICEX box in our study and due to the high spatial and temporal coverage of PIOMAS. We have however tempered our statements to reflect the uncertainty of PIOMAS.**

**Given the uncertainties of both observational products and PIOMAS, using all of the products together allows us to obtain a range of 'reference values' that is more reliable than the range based on observational products alone.**

**For drift speed, since we now include the merged product from Tschudi et al. (2016) and due to the temporal resolution of PIOMAS sea ice velocity vectors (monthly), we decided to remove PIOMAS drift speed from our analysis.**

- p. 5 l. 20: I guess the paragraph on p. 4 l. 29 belongs here. Just keep in mind that even though Schweiger et al. (2011) is a very nice paper, then PIOMAS is not the truth. I would ask you to reduce considerably your reliance on PIOMAS in this study and try to compare to actual observations instead, as flawed as they may be. You also haven't considered the Rothrock et al. (2008) multiple regression model, which is well worth taking into account here.

**Based on the comments from both reviewers regarding PIOMAS, we now use ULS submarine ice thickness observations as well as the drift speed merged product of Tschudi et al. (2016) in our study, and we removed PIOMAS drift speed. Please see our response to the previous comment related to the criticism of PIOMAS.**

- p. 6 l. 6: This is not the right reasoning for choosing daily time scales. With daily time scales you capture synoptic-scale variability, but with monthly time scales you average these out and capture the longer-term, large-scale drift.

**We rephrased according to the reviewer's suggestion.**

- p. 6 l. 8: Given my comment above it should be clear that you cannot use the monthly values from PIOMAS in this way. They contain different physics and you can't just scale with factor two!

**We decided to remove PIOMAS drift speed from our analysis due to the error linked to this scaling and the poor results obtained with this reanalysis in terms of drift speed.**

**According to our results with NEMO-LIM3.6, the scaling between monthly sea ice drift speed derived from daily components of velocity and the one derived from monthly components is two (Fig. 2 in the revised manuscript). A recent study focussing on Arctic sea ice drift speed using 22 CMIP5 models shows that this factor 2 is valid for all the models (Tandon et al., submitted). Therefore, we think that it is a good approximation, even if we agree that there is an uncertainty linked to this scaling.**

- p. 6 l. 21: From here on out this section becomes increasingly hard to understand. I had to re-read and then re-read again to completely understand which metrics and diagnostics you use. It's all there, but you're making your reader work way too hard to get the point. Please rewrite and try to make it clearer and better organised.

**We re-organised this part of the text to make it clearer.**

- p. 6 l. 28: The novelty here is really that you use this method as a way to evaluate your model.

**We removed this sentence after the re-organisation made in response to the previous comment.**

- p. 6 l. 29: You don't normalise with wind friction speed, but Olason and Notz (2014) say they do this to take atmospheric stability into account. It is interesting that you find that this is not necessary, it is not what they find. However, I don't understand why you don't normalise with the 10 m wind speed at least, since we know there should be a close correlation between drift speed and wind speed. My main concern, however, is that your figure 9b gives a completely different shape for the curve than figure 6 from Olason and Notz (2014). Why is that?

**When normalising sea ice drift speed by wind friction speed (Fig. R2 below), we obtain very similar drift-concentration and drift-thickness relationships compared to these relationships without normalisation (compare Fig. R2 below to Fig. 8a-b in the revised manuscript). That is why we decide not to normalise. We think it is easier to interpret (physically speaking) direct data instead of normalised data. Please see also our response to the specific comment of Referee #2 (page 9, line 16-17).**

[Figure]

*Fig. R2: Scatter plots of modelled (NEMO-LIM3.6) and observed monthly mean normalised sea ice drift speed against (a) concentration and (b) thickness spatially averaged over the SCICEX box and temporally averaged over the period 1979-2013 (except when stipulated in the legend). Drift speed is normalised by wind friction speed (derived from DFS5.2) as in Olason and Notz (2014).*

**We do not think we obtain a completely different shape for the observed drift-thickness relationship compared to Olason and Notz (2014). Please see Fig. R2b above, where our black curve (IABP/ULS) is very similar to the blue curve of Fig. 6b in Olason and Notz (2014). Our blue curve (IABP/PIOMAS) in Fig. R2b shows thinner ice compared to the red curve of Fig. 6b in Olason and Notz (2014) but we think that is mainly due to the period used, i.e. 1979-2013 in our study and probably 1979-2000 in Olason and Notz (2014) in order to compare to ULS observations. We demonstrate this effect of the period by also plotting sea ice thickness from PIOMAS averaged over 1979-2000 (see gray dashed curve in Fig. R2b): the resulting curve is much closer to the red curve of Fig. 6b in Olason and Notz (2014). The remaining differences probably arise from the slightly different period: Olason and Notz (2014) do not say over which period they compute the mean seasonal cycle in their Fig. 6b.**

- p. 7 l. 9: These are probably good metrics you've developed, but you don't use them enough and you don't discuss them enough to make me want to use them too.

**We make use of these metrics in Sections 3.2 and 3.3 as well as in Figs. 8 and 13 in the revised manuscript. We also discuss the use of process-based metrics in the framework of a model intercomparison in Section 4.4.**

- p. 7 l. 15: Mention (again) the period you average over.

**Done.**

- p. 7 l. 16: What are the (main) differences between your set up and Rousset's et al. (2015)? If it's just the resolution then remind the reader which resolution you use.

**The resolution is different (1° for our study and 2° for Rousset et al. [2015]) as well as the atmospheric forcing (DFS5.2 for our study and CORE normal year for Rousset et al. [2015]). We added this detail in the text.**

- p. 7 l. 27 (all paragraph): I'm concerned that you rely too much on comparison with PIOMAS. Again, it's only a model so you should try hard(er) to compare to observations before resorting to comparing with PIOMAS.

**We considered the remarks from both reviewers by using other observational datasets. Please see our response to the comment 'p. 4 l. 29'. Furthermore, we also compare the modelled sea ice thickness to ICESat in this paragraph. However, as explained before, the temporal coverage of the latter dataset is very limited (2003-2008) and the measurements suffer from uncertainties.**

- p. 8 l. 6 (the paragraph and this section in general) You jump a lot between the SCICEX box and your "wider domain" and I'm having trouble keeping up. Try to decide which is more important, stick to it and mention the other one only when necessary.

**We removed all results related to the wider domain from Section 3 (Results) and synthesised this information in Section 4.3 (Impact of domain choice) to make it less confusing.**

- p.8 l. 26: What conclusion should I draw from this paragraph? Is the trend significant or a post-processing glitch?

**We rephrased to make it clearer. The main conclusions are that modelled trends are good for sea ice concentration and thickness, are less good for sea ice extent (model underestimation, especially in winter) and do not capture the observed positive summer trends in drift speed provided by IABP buoys.**

- p. 10 l. 6: You need a justification for using lambda and not tuning P*

**We now use P\* experiments. Please see our response to a previous comment (p. 3 l. 29-32).**

- p. 10 l. 14: It's not counter-intuitive to me, as I mentioned earlier when commenting on the abstract.

**We removed this sentence following our results explained above (comment p. 1 l. 11).**

- p. 10 l. 25: It's not really a hysteresis loop. Physically the drift speed depends on ice thickness only when the concentration is high, so the change in drift speed only relates to the change in thickness in winter.

**We think it is a hysteresis loop in the sense that for a given sea ice thickness, two different drift speed values are found for a given thickness depending on the season (summer vs. winter). As we show in Fig. 8b in the revised manuscript, drift speed does not only depend on thickness when concentration is high (October to March) but also when concentration is low (May to September).**

- p. 11 l. 2: Your heat-flux theory contradicts the results of Steel et al. and Flato and Hibler. You need to show that it's true by actually showing the ocean-atmosphere heat flux and analysing that.

**We found that our 'heat-flux theory' was not confirmed by the results obtained with the P\* experiments (see our response to comment p. 1 l. 11). The manuscript has been revised accordingly.**

- p. 11 l. 22: It's only a physical correlation in winter. See my comment for p. 10 l. 25

**We do not have enough arguments to make such a statement (only physical correlation in winter). What we observe when we plot drift speed against thickness is an anti-correlation between both variables in both summer and winter. This does not infer causality.**

- p. 11 l. 28: I don't know what you mean by "large-scale effects"

**This has been removed.**

- p. 13 l. 16: I can draw no concrete conclusions from this sub-section

**This has been revised. Please see also our response to a previous comment (p. 8 l. 6).**

- Figures 5, 6, 8, and 12: These figures are very small and hard to read. It would be better if they focused on the central Arctic. I don't think you would lose much information doing that. Figure 6 is particularly hard to read and I can't see the directions of the vectors at all. It would also be nicer to have the magnitude and direction of the difference, rather than the way it's done now.

**Done.**

---

## Author Comment (AC2) · 13 Jul 2017

**Reply to Referee #2**

**Interactions between Arctic sea ice drift and strength modelled by NEMO-LIM3.6**

**Docquier *et al.* (2017), tc-2017-60**

We would like to thank Referee #2 for his/her very constructive feedback, which has helped us improve the paper quality. Below we present our detailed responses to the comments and suggestions proposed by the reviewer in **red**. The corresponding corrections are in **red** in the revised manuscript.

**1. Summary**

The authors conducted ocean-sea ice model NEMO-LIM3.6 numerical experiments to investigate interaction and/or feedback mechanisms between sea ice drift speed and ice strength, focusing on the Arctic Ocean. As measures of ice strength, the authors employ ice concentration and thickness, and then examined relation between ice drift speed and them. In order to assess the model performance, sea ice observation data derived from satellite and ice-tethered buoys (ice concentration, ice drift speed and ice thickness) were exploited. The authors introduced a systematic model validation method based on sea ice diagnostics (ice extent, ice concentration, ice thickness and ice drift speed) as well as process-based diagnostics (relation between different ice properties, e.g., ice drift speed and concentration). They also introduced metrics which quantify modeled or observed ice dynamics as scalar variables. Using these diagnostic and metric approaches, the author assessed model performance, and then conducted sensitivity experiments varying ice strength parameter to see the effect of ice strength change on the relation between drift speed and ice concentration (or ice thickness) simulated in the model.

**2. General comments**

The authors provided a good review regarding relation between ice drift speed and ice concentration (and thickness) in the Arctic Ocean, mainly focusing on observational studies. The diagnostics and metrics used to validate model performance and the thorough assessments on the simulated ice properties are very helpful to assess the capability and limitation of the model, although I have an additional suggestion regarding ice thickness evaluation (see comments below). The results of the sensitivity experiment, in which ice strength parameterization is changed, are interesting to me (and probably informative to many sea ice modellers), while the whole strategy used to assess the feedback between ice drift and other ice properties seems to me not always suitable nor convincing for the purpose of the study. As far as I know, "feedback" is a term originally introduced to describe an electric circuit which has a recursive input, and is widely used in other study area to explain similar concept. Also in geo-scientific studies, the term "feedback" is widely used to explain a recursive interaction between different processes or subsequent chain of phenomena, sometimes without clear definition nor quantification. Although the authors invoked the term "feedback", the strategy does not seem to be designed so as to extract a feedback mechanism from a complicated system. For this reason, I would recommend to change the focus of the study to more specific issues (e.g.,

sensitivity of ice strength parameterization to ice drift, concentration and thickness) or to reorganize experiment design so as to quantify the feedback between them. Even without feedback issue, the manuscript contains interesting model results.

**We thank the reviewer for his comments. Since his point related to the feedback methodology is related to the first major point below, we provide our answer there.**

**3. Major points**

1) The strategy used to extract (and quantify) the drift-strength feedback is not always suitable nor convincing. I think more consideration is necessary about how to quantify a "feedback". Since ice drift, concentration and thickness are described by a set of simultaneous partial differential equations in numerical models, it is a matter of course that there is some sort of 'feedback' between them. An important point is how to extract and quantify a feedback in a simple formula so as to abstract the essence of the complicated system. Mathematically, a feedback between two variables (most simple case) can be described by a set of equations,

$$\frac{d\overline{|V|}}{dt} = F(\overline{T}) \tag{1}$$

$$\frac{d\overline{T}}{dt} = G(\overline{|V|}) \tag{2}$$

where |V| and T are respectively mean ice drift velocity and ice strength (or thickness) in the present case (e.g., |V| = 1/TS ∫∫ |V|dxdydt , T: one month or one year period, S: SCICEX box). F and G are the functions describing a feedback. The equations mean that the temporal evolution of mean ice drift is controlled (or affected) by mean ice strength, while at the same time, the evolution of ice strength is also controlled by drift speed. The most simple solution is an exponential formula, $Ce^{\alpha t}$, and then temporal evolution of |V| and T depends on α. The system has a positive feedback if α > 0, while a negative feedback if α < 0. If the authors intend to clarify and quantify the feedback mechanism between ice drift and thickness, a definite formula (not necessarily means complicated formula) in such a simple theoretical framework should be provided. Otherwise, we cannot learn anything about the quantitative features of the feedback between ice drift and thickness (or concentration) at all. Note that the system also needs a higher order stabilizing term to prevent exponential growth of a positive feedback. (At least discussion for such a damping mechanism is necessary.)

**We agree with the reviewer that our study probably lacks a robust mathematical framework to quantify the sea ice drift-strength feedback. Due to the complexity of this feedback (see sensitivity experiments) and the lack of observations confirming the existence of this feedback in real life (see main reservation of Referee #1), we decided to change the focus of this study by analysing the interactions between sea ice drift and strength (concentration and thickness) rather than the feedback itself. The main results stay similar but the wording is now slightly different in the revised manuscript: we talk more about interactions between drift speed and strength and about the impact of changes in strength parameterisation on the resulting drift speed and thickness, rather than about the feedback itself. We also changed the article title accordingly.**

2) The design of the sensitivity experiment seems to me not suitable for examining the effect of ice strength change on ice drift, ice thickness and concentration. Since the authors change the ice strength, P, by changing exponent of ice thickness h, the associated change of ice strength has an opposite sign between h > 1 and h < 1. This is easily confirmed by calculating P value for different λ used in the sensitivity experiment: if h > 1, P λ=2 is larger than P λ=1 , while P λ=2 is smaller than P λ=1 if h < 1 (this is also confirmed by closely looking Fig. 2). Due to this experiment design, we cannot directly relate the change of parameter λ to the change of ice strength and therefore, to changes of the ice thickness and ice concentration, which makes interpretation of the results complicated and difficult (There are many misinterpretation in sec. 3.3 due to this fact, see the specific points below). Generally speaking, changing exponent of h in the ice strength equation is not equivalent to changing ice strength, but to changing sensitivity of ice strength to h. My recommendation is to use more simple formula for the sensitivity experiment (e.g., P = P* λh exp[-C (1 - A)] , which is equivalent to change P*), or to redo analyses based on the actual ice strength P used in the model (i.e., calculate P= 1/TS ∫∫ P( x , y , t) dxdydt and examine relation between P and modeled ice properties; ice drift, ice concentration and thickness).

**We agree with the reviewer that an increase (decrease) in λ for thickness h < 1 m leads to smaller (higher) ice strength P. Due to this problem and the remarks of both reviewers regarding the design of our sensitivity experiments, we performed new sensitivity experiments in which we change the P\* parameter according to values found in the literature. The chosen P\* values are described in our response to Referee #1 (see his minor comment p. 3 l. 29-32) and a detailed explanation of the P\* experiments has been added to the manuscript (Section 2.1). Results are presented in Section 3.3 and lead to the same conclusions as for λ experiments, i.e. a higher (smaller) initial ice strength, caused by higher (smaller) P\*, leads to lower (higher) sea ice thickness.**

**In the revised manuscript, we decided not to show the results from λ experiments but we discuss them in Section 4.2 as they are still interesting for several reasons:**

   a. **Some sea ice models use P scaling as h[1.5] (e.g. Lipscomb et al., 2007) since it is more physically realistic than P scaling as h (but less numerically stable); therefore it makes sense to test the sensitivity of the model to different λ values.**
   b. **Even if it is true that P decreases with increasing λ for h < 1 m, the differences in P between the four different experiments for h < 1 m are very small; these differences become more important for h > 2 m as shown in Fig. 2 in the previous version of the manuscript; moreover, the mean ice thickness averaged over the SCICEX box (the quantity we use in our study) is always higher than 1 m, even in September when ice is the thinnest (Fig. 10a in the previous version of the manuscript).**
   c. **These experiments are original (most modelling studies change P\*) and this is the first time that four λ values are used (the existing studies only look at λ = 1 and λ = 2).**
   d. **The λ experiments are more complex than the P\* experiments (due to their exponential nature), but they confirm the results obtained with the latter experiments.**
   e. **Leppäranta (2011) states that the value of λ is an open question.**

3) I admire the systematic approach used in this study to assess the model performance using observational data, whereas I have a concern about the use of ice drift and thickness data derived from PIOMAS, instead of direct observations. The authors did not utilize two important dataset for ice drift and thickness, which have sufficient spatial and temporal coverage for the present study. One is ice drift data provided from Colorado University group (Tschudi et al., 2016), the other is the long-term ice thickness estimate by Lindsay and Schweiger (2016). Both estimates provide error of the estimates as well. Since these data were derived from in-situ and satellite measurements while PIOMAS did not assimilate ice drift and thickness, I recommend to use these two data instead of PIOMAS simulation. Note that the bias (or error) of ice drift field in Tschudi et al. (2016) reported by Szanyi et al. (2016) is a crucial issue, only if the data is used for divergence/convergence calculation. Since the divergence/convergence features reported by Szanyi et al. (2016) always appear as a divergence/convergence pair around buoy-data merged location, a spatial averaging (e.g., SCICEX box) can eliminate the error.

**Following the remarks related to PIOMAS from both reviewers, we now use the multiple regression model from Rothrock et al. (2008) based on ULS submarine ice draft measurements to derive sea ice thickness. These data constitute the longest record in terms of sea ice thickness (1975-2000), they cover the SCICEX box that we mainly use in our study and they are integrated into the Lindsay and Schweiger (2015) dataset.**

**For the observed drift speed, we now include the NSIDC merged dataset from Tschudi et al. (2016), which includes IABP buoy data. We call this dataset 'NSIDC' throughout the paper. However, we are a bit skeptical about the mean seasonal cycle derived from these data (Fig. 3d in the revised manuscript), which gives drift speed values up to 4 km/d lower than the IABP buoy drift speed values. The amplitude of the seasonal cycle of the merged product is also much lower than the one from the buoy data. These features were already noticed by Olason and Notz (2014) who used the previous version of the NSIDC merged product (Fowler et al., 2013) (see their Fig. 10).**

**Please note that we already use IABP buoy data from the Tschudi et al. (2016) dataset as well as OSI SAF data for drift speed, and ICESat for ice thickness. We decided to remove PIOMAS drift speed from our analysis due to the poor results associated with this reanalysis for drift speed as well as the scaling problem (factor 2).**

**4. Specific points / additional comments**

- Page 2, line 1-8: I would suggest the author to use the term 'feedback' more carefully, particularly when mentioning 'positive feedback'. Since a positive feedback has an exponential growth feature by its definition, it always needs damping or stabilizing mechanism in higher order, when the concept is used to explain things occurring in the nature.

**Please see our response to the first major point above.**

- Page 2, line 9-22: I appreciate the good summary of the former studies here, which is useful to survey the current status of our understanding on this issue. On the other hand, I am a little bit skeptical to provide conceptual illustration like Fig. 1, since such an illustration sometimes goes out of authors' control if once published, even if each arrow in the figure is not really examined.

**We removed the diagram (Fig. 1 in the previous version of the manuscript) due to our decision not to focus on the feedback but rather on interactions (see our response to the first major point).**

- Page 3, line 29-30: Why the authors conducted the sensitivity experiment by changing exponent of h in the ice strength equation, instead of changing P* or C? If the authors intend to increase ice strength in the entire h range, changing P* seems to be more suitable approach (as many modellers do). I request more explanation.

**As mentioned in our response to the second major point, we ran new P\* experiments and we decided not to show results from λ experiments (but we discuss them in Section 4.2). Finally, we had already carried out experiments by varying C between 16 and 22, but we found negligible differences in sea ice thickness and drift speed, so we decided not to include them in our study.**

- Page 4, line 1-7: Since we cannot directly relate the increase of λ to increase of ice strength in the present experiment design, more careful explanation is needed. For example, ice thickness during the summer season in the SCICEX box may thinner than 1 m, at least part of the area. In this case, an increase of λ leads to decrease of ice strength.

**Please see our response to the second major point above.**

- Page 4, line 26-34: Why the authors did apply ice drift data provided from PIOMAS instead of satellite- and buoy-based data as a long-term ice drift observation? Polar Pathfinder Daily 25 km EASE-Grid Sea Ice Motion Vectors (Tschudi et al., 2016, now version 3 is available via NSIDC website) provides Arctic-wide long time series from 1979 to present. Although PIOMAS reasonably reproduced sea ice extent and ice concentration field as a result of ice concentration assimilation, there is no reason to believe that ice drift and thickness from PIOMAS are consistent with observation, since these variables are not assimilated. I think the ice drift data from PIOMAS should be dealt with a great care (Since PIOMAS applies strong constraint on ice concentration by an optimal interpolation, the simulated ice field never breaks down, even if ice drift is totally unrealistic). Use of Polar Pathfinder data is much more reliable approach, since the uncertainty of the estimates are also provided. The problem of the Polar Pathfinder data reported by Szanyi et al. (2016) is not a critical issue for the present application (see my major point as well).

**We removed PIOMAS drift speed from our analysis and now use the NSIDC dataset from Tschudi et al. (2016).**

- Page 5, line 20-24: For long-term ice thickness estimates in the Arctic Ocean, I strongly recommend to use the estimate presented in Lindsay and Schweiger [2015]. They provided an empirical function describing the spatially and temporally varying ice thickness field over the Arctic Ocean, by exploiting all available in-situ and satellite measurements of ice thickness. As far as I know this is the most reliable long-term estimate of ice thickness field based on measurement for the time being (One can calculate seasonally varying long-term ice thickness field by the description in Lindsay and Schweiger).

**We are aware of the sea ice thickness dataset from Lindsay and Schweiger (2015). However, we decided to use only one source of data, namely ULS submarines, due to the fact that these data cover the SCICEX box (which we use as a region of interest in our study) with sufficient accuracy.**

- Page 6, line 1-4: How did the authors define the daily ice drift in Eq. (2)?

$$u_d = \frac{1}{T_{day}} \int_{t_1}^{t_2} v(t)\,dt \qquad \text{(i.e., temporal average of instant velocity in x and y direction)}$$

or $\qquad u_d = \frac{\left| x_{t_2} - x_{t_1} \right|}{T_{day}} \qquad$ (zonal or meridional displacement for 24 hours)?

The definition of satellite- or buoy-derived ice drift is the latter. I don't think the difference between the two definitions is large, but it would be nice if the authors clarify their definition for comparison with other model results.

**For our model, daily mean ice drift speed is defined by Eq. (2), where $u_d$ and $v_d$ are daily mean ice velocity components. We articulate this in the revised manuscript. $u_d$ and $v_d$ are computed in the model by averaging values over the 8 model time steps covering each day, i.e. following the first equation mentioned by the reviewer above.**

- Page 7, line 27-30, Page 8, line 1-5: I think the comparison with PIOMAS thickness data provides useful insights, while I am skeptical to regard PIOMAS thickness data as substitution for observation, since we cannot distinguish bias or error of the estimates, which are always provided for observational data.

**We added observations from ULS submarines. However, we also think that PIOMAS is useful since its thickness is realistic compared to observations (Schweiger et al., 2011) and due to high uncertainties related to ice thickness observations. Please also see our response to the third major point above as well as our response to the comment 'p. 4 l. 29' of Referee #1.**

- Page 8, line 1: Do the authors have an explanation why the peak shifts?

**Modelled sea ice thickness averaged over the SCICEX box and over the domain north of 50°N is shown below (Fig. S1 below). As shown, the SCICEX maximum occurs in May while the 50°N maximum occurs in June with a second peak in August. We do not have an explanation for this shift. However, the 50°N is a very large domain that takes into account coastal areas, so we are a bit skeptical of the representativeness of the 50°N seasonal cycle.**

[Figure]

*Fig. S1: Modelled (NEMO-LIM3.6) monthly mean seasonal cycle of Arctic sea ice thickness (sea ice volume per area) temporally averaged over the period 1979-2013 for two different domains (solid line: SCICEX box; dashed line: north of 50°N with A >= 0.15).*

- Page 8, line 23-24: How the IABP ice drift data were processed to calculate spatial (and temporal) average over the SCICEX box? Since the spatial coverage of the buoy data is not sufficient, one needs to interpolate/extrapolate the data. How did the author define influential radius of the buoy data? How much uncertainty should we expect from the interpolation/extrapolation process? Since the IABP data averaged over the SCICEX box is an important measure to validate the model, a description is necessary.

**For IABP drift speed data, we first computed daily spatial means by taking all buoys within the SCICEX box. Then, we computed monthly means from these results. By doing so, we do not need to interpolate / extrapolate and there is no error due to the spatial sampling. The spatial coverage of buoy data within the SCICEX box is fairly good (Fig. S2 below, coming from Rampal et al. [2016], Fig. 2). However, we acknowledge that comparing buoy and model drift speed has to be done with caution, as buoys measure the drift speed at one particular location while the model is meant to give the grid-cell average. This information has been added in the manuscript (Section 2.3).**

[Figure]

*Fig. S2: Buoy tracks from the IABP dataset for the winter periods 1979–2011 (left panel) and the corresponding number of buoys (middle panel) and records (right panel). Source: Fig. 2 from Rampal et al. (2016).*

- Page 9, line 2-3: Why the authors show the relationships in terms of mean seasonal cycle, not by scatter plots based on the relations in each month? I mean, for example, a scatter plot for drift-concentration relation in each month can more clearly show the validity of the regression line.

**Our aim is to show how changes in drift speed are linked to changes in concentration and thickness for each month of the year (mean seasonal cycle) for both the model and the observations as in Olason and Notz (2014). We agree with the reviewer that it is also interesting to show these relationships for each month of every year but this produces a noisier picture (especially for the thickness loop). Please see Fig. S3 below, where we plot scatter plots for every month of each year. They show that drift-concentration and drift-thickness slopes are similar to the ones using the mean seasonal cycle (Fig. 8a-b in the revised manuscript), with slightly lower model performance. This information has been added in the revised manuscript.**

[Figure]

*Fig. S3: Scatter plots of modelled (NEMO-LIM3.6) and observed monthly mean sea ice drift speed against (a) concentration and (b) thickness spatially averaged over the SCICEX box for each month of the period 1979-2013. Slope ratios and normalised distances between NEMO-LIM3.6 and the different observation/reanalysis datasets are shown in brackets in the legends.*

- Page 9, line 16-17: Why the authors did not apply normalization by wind stress? Since the wind stress may differ between each month, it is difficult to derive a general relation between drift speed and other variables. If there is a reasonable explanation for not to apply normalization, please describe in the text.

**For normalising drift speed by wind friction speed, we use the same methodology as in Olason and Notz (2014), except that we use wind speed from DFS5.2 since this is the atmospheric forcing of our model. We also use air density of 1.225 kg/m³ and drag coefficient of 0.0015. As shown in Fig. S4 below, the normalised drift-concentration and normalised drift-thickness relationships look very similar to drift-concentration and drift-thickness relationships without normalisation (compare to Fig. 8a-b in the revised manuscript). Therefore, we decided not to perform the normalisation, which facilitates the interpretation of data. Please see Section 2.3 of the manuscript: 'A key difference with Olason and Notz (2014) is that we do not normalise drift speed by wind friction**

**speed since our findings were not sensitive to such normalisation'. Please also see our response to Referee #1 (p. 6 l. 29).**

[Figure]

*Fig. S4: Scatter plots of modelled (NEMO-LIM3.6) and observed monthly mean normalised sea ice drift speed against (a) concentration and (b) thickness spatially averaged over the SCICEX box and temporally averaged over the period 1979-2013. Drift speed is normalised by wind friction speed (derived from DFS5.2) as in Olason and Notz (2014).*

- Page 9, line 30-31: I think the use of 'feedback' in this sentence is not appropriate. The result in this section (sec. 3.2) shows that there is a (linear) relationship between drift speed and strength (thickness or concentration) as equilibrium states (A feedback system does not reach an equilibrium state unless α = 0, or having oscillating solution. see my major point). I don't mean the analysis in this section is meaningless, but is not appropriate to show the existence of feedback (see also major point).

**We changed the term 'feedback' into 'interactions' as we think it is more appropriate. Please also see our response to the first major point above.**

- Page 10, line 9-20: There are a number of incorrect sentences here. Please keep in mind that higher λ does not directly correspond to larger ice strength, due to the current formulation, Eq. (1). Particularly, it is not true, when discussing relation between ice drift and thickness (or concentration) in summer season. The thickness may thinner than 1 m at least part of the SCICEX box.

**This part has been re-written due to the inclusion of new P\* experiments and in response to the second major point of the reviewer.**

**The mean seasonal cycle of ice strength P for the different λ values is shown in Fig. S5a below, where we see that ice strength is higher for higher λ most of the time, with very small differences in summer compared to winter. The zoom in summer months (Fig. S5b below) reveals that this relationship still holds (higher strength for higher λ), except for the λ = 2 curve in summer, which is located between λ = 1 and λ = 1.5 in July, and between λ = 0.5 and λ = 1 in August and September.**

[Figure]

*Fig. S5: (a) Modelled (NEMO-LIM3.6) monthly mean seasonal cycles of sea ice strength temporally averaged over the period 1979-2013 and spatially averaged over the SCICEX box for four different λ values (see Eq. (1)). (b) Snapshot of (a) for June-September.*

- Page 10, line 16-20: I think this interpretation is wrong, probably due to the fact which I described in major point. Since the ice thickness during summer season is close to (or even smaller than 1 m), the ice strength for larger λ becomes smaller than that for smaller λ. Therefore, the larger drift speed for larger λ can be simply the result of smaller ice strength.

**As demonstrated in our response to the previous comment and in Fig. S5, higher λ leads to higher strength when averaging over the SCICEX box, except for the λ = 2 curve from July to September. Therefore, the larger drift speed for larger λ in summer in Fig. 10b in the previous version of the manuscript is not driven by lower ice strength P (except maybe for λ = 2). Furthermore, a similar behaviour is observed with P\* experiments: higher P\* leads to higher drift speed in summer (except P\* = 100kN/m²) (Fig. 9b in the revised manuscript).**

- Page 10, line 32-33: I would say the result is not counter-intuitive. Note that increase of λ leads to smaller ice strength in h < 1 area, it means ice can be easily deformed or compressed in h < 1 area, compared to small λ.

**We rephrased this following comments from both reviewers and due to the use of new P\* experiments.**

- Page 10, line 4 - page 11, line 8: Section 3.3 needs additional figure showing the relation between λ and mean ice strength in SCICEX box (a seasonal cycle for each λ should be shown), otherwise, we cannot relate ice strength with ice thickness (and concentration) nor examine the relation between ice strength and ice drift.

**Please see our response to the comment 'Page 10, line 9-20' and Fig. S5. However, the main focus is now on P\* experiments, which are simpler to interpret. We only discuss λ experiments in Section 4.2.**

- Page 10, line 33 - page 11 line 6: I think the reason for the increase of modal thickness for larger λ (Fig. 13) can be simply explained. The experiment with increased λ has larger ice strength for h > 1 (winter season), which prevents ice thickening due to ridging, while it has smaller ice strength for h < 1 (summer season), which enhance ice thickening due to ridging. As a result, larger λ leads to larger peak at modal thickness.

**We are not sure about the validity of this hypothesis since we obtain similar results with the new P\* experiments (in which larger P\* always lead to larger ice strength P), i.e. decrease of modal thickness and decrease of ice thickness heterogeneity for larger P\* (Fig. 12 in the revised manuscript).**

- Page 11, line 14-15: I would say that this study also did not 'quantify' the magnitude of the drift-strength feedback. To quantify a feedback, one should present growth rate of the feedback.

**We rephrased this paragraph since we focus now on the interactions (and not on the feedback) between drift speed, concentration and thickness. Please see our response to the first major point above.**

- Page 11, line 23-24: Due to the analyses without normalization by wind stress, it is difficult to distinguish the reason for the hysteresis loop shown in Fig. 9b and Fig. 11b. Can the hysteresis loop be observed if the ice drift speed is normalized by wind stress?

**As shown in Fig. S4b, a hysteresis loop is also present when normalising drift speed by wind friction speed.**

- Page 12, line 14 - page 13, line 15: As pointed in major point, I think the basic strategy for the sensitivity experiment and analyses are not suitable for quantifying 'feedback'. If the authors intend to quantify 'feedback', the entire framework should be reconsidered.

**Please see our response to the first major point above.**

- Page 14, line 16-18: Why such hysteresis loops are observed in the modeled sea ice? Since observation (IABP) does not show such a feature, I guess this is an artifact coming from insufficient modeled physics.

**As shown in Fig. 8b in the revised manuscript and in Fig. 4b of Olason and Notz (2014), observed drift-thickness relationships are marked by a hysteresis loop: for a given thickness, two different drift speed values exist depending on the season.**

- Page 14, line 21-28: The summary provided here should be reconsidered, by taking the second paragraph of the major point into account. Although the authors generally provided descriptive result of their analyses on ice drift - thickness relations, they did not show any results on thermodynamic analyses. Therefore I cannot follow nor rely on the arguments associated with thermodynamic effects.

**We re-wrote this part of the text since a new set of P\* experiments have been performed. Concerning the thermodynamic analyses, we refer the reviewer to our response to the first minor comment of Referee #1 (p. 1 l. 11).**

**5. References**

- Lindsay, R. and A. Schweiger, 2015: Arctic sea ice thickness loss determined using subsurface, aircraft, and satellite observations, The Cryosphere, 9, 269-283, doi:10.5194/tc-9-269-2015.

- Szanyi, S., J. V. Lukovich, D. G. Barber, and G. Haller, 2016: Persistent artifacts in the NSIDC ice motion data set and their implications for analysis, Geophys. Res. Lett., pp. 10800-10807, doi:10.1002/2016GL069799.

- Tschudi, M., C. Fowler, J. Maslanik, J. S. Stewart, and W. Meier, 2016: Polar Pathfinder Daily 25 km EASE-Grid Sea Ice Motion Vectors. Version 3. Boulder, Colorado, USA. (https://nsidc.org/data/docs/daac/nsidc0116_icemotion.gd.html).

**Additional references included in this reply that are not in the revised manuscript:**

- **Fowler, C., W. Emery, and M. Tschudi, 2013: Polar Pathfinder Daily 25 km EASE-Grid Sea Ice Motion Vectors, version 2, NSIDC, Boulder, Colorado.**

---

## Referee Report (RR1)

**Review of "Interactions between Arctic sea ice drift and strength modelled by NEMO-LIM3.6"**

I would like to thank the authors for their responses to my previous remarks. It is clear that they took my remarks under serious consideration. They have, in particular restructured and refocused the article as, in fact both I and the other reviewer felt was necessary. As it stands, the paper is much closer to be acceptable for publication, than was the original submission.

Despite the good progress made I still have two major and some minor remarks that I would like the authors to respond to, before I can recommend publication.

**Major comments**

My first major comment is about the drift vs. thickness plots. Reading Olason and Notz, it is my understanding that they view this relationship to be only of relevance during winter, i.e. from November to March. And I agree. The reason for this is that it is only in winter that the ice cover is compact enough so that the ice thickness can play a role in the momentum equation. If the ice cover is very loose the open water part of the cover will "deform", so to speak, regardless of the thickness of the ice. The straight line between November and March is thus physical, while the straight line between May and September is coincidental.

You can aso see this when looking at the IABP vs PIOMAS curve. I know I said "PIOMAS is not observations" last time, but we know that the shape of the Rothrock et al curve is wrong (they say as much in the paper themselves), and it's reasonable to assume that PIOMAS is closer to reality in that regard. In IABP vs PIOMAS the straight line in summer is only between July and September; it's still a straight line, but I would argue that this is coincidental.

Keeping this in mind your metrics should refer only to the November-March period, but it seems the entire curve is taken into account. This is crucial. Also, if you did this you could use your figure 13 b) to see that w.r.t. drift speed $P^* = 27.5$ gives (close to) the right slope, i.e. drift speed-thickness relationship.

My second major comment is that I feel that the structure of the paper can still, relatively easily be improved. I will not stand particularly firm on this, it is more a stylistic comment than a strictly scientific one. But I honestly believe that by improving the structure you can make the paper better focused on the important new methods and results, which will help in making it more widely read, recognised, and cited.

I suggest you really focus on the new metrics you developed. You would then first introduce the metrics and then evaluate NEMO-LIM using them. This second step shows the applicability of the metrics and evaluates the model at the same time. You can then use the

P* experiments to show how you can use the metrics to aid with sensitivity studies. A plot of s_h and epsilon_h against P* w. Finally you show, through the P* experiments, that the drift speed in summer doesn't depend on ice thickness.

This way you put the new metrics and new science in foreground and the evaluation of NEMO-LIM in the background. That would, in my opinion, change the paper from a somewhat utilitarian model evaluation paper to one presenting new metrics and new findings.

**Minor comments**

P.2 L.5: Replace "At large scale" with "To first order", or something similar. Changes in ice strength, unrelated to concentration and thickness can have a large scale effect (see Girard et al and the work that follows them).

P.2 L.11: You introduce the phrase "drift-strength feedback", but don't use it again in the revised paper. This sentence is not needed. Also I thought we agreed not to say "feedback" :)

P.2 L.16: I would like you to say "most likely caused by reduced thickness and concentration".

P.2 L.32: change to "... multi-model dataset suggests that in those models thicker and more packed …"

P3. L.18: Change the subsection title to something along the lines of "Model and sensitivity experiment description". Now it sounds like you'll describe the model and perform the sensitivity experiments, but you're in fact describing both.

P6. L.29: You added "sea ice area", but do you really use that? Also, how does that differ from sea ice concentration. One is the true area and the other the fractional area, right?

P7. L.4: You discuss the monthly mean drift speed computed from the daily components of ice velocity at length here. But I don't think it's needed. You don't use this information in the rest of the paper and it's not really relevant to what you're presenting here. Please remove this discussion.

P.7 L.31: When introducing the metrics you don't say which months you use to calculate them. For the concentration related metrics this is probably June-October already(?), but for the thickness it should be November-March, as I mention in the major comments.

P.8 L.12: the part of "which is based on a combination … on the mean 1984-2000 seasonal cycle" is not really necessary here.

P.9 L.10: You should show the OSI SAF drift, even if it doesn't cover summer. There is a huge discrepancy between the two observations you do show, so throwing in the third gives us a better idea of how serious this discrepancy is. Also, the NSIDC data is not really good in summer, last I checked.

P.10 L.5: (Paragraph) See my major point about drift-thickness relationship being coincidental in summer.

P10. L.13: (Paragraph) I would skip this paragraph and figure. It's not a bad idea, but you need to dedicate much more time/space to discuss the physics and observations, since you have no one to cite on this. For figure 8c: This makes sense, since you'd expect a trend in concentration to coincide with a trend in speed, but the model does the oposite (which is interesting). Why don't you show the NSIDC speeds here? For figure 8d: I can't read anything useful out of this one: A trend in thickness coincides with a trend in drift speed mainly in months where there is no relationship between drift speed and thickness.

P.10 L.23: Skip this last line as it refers to the paragraph I recommend skipping above.

P.11 L.15: (Paragraph) If you just consider the winter months (which is where changing P* really has han effect) then you see that P* = 45 kN/m^2 is actually quite a good fit.

P.11 L.20: (Paragraph) If you consider only November-March for the drift-thickness relationship then P* = 27.5 kN/m^2 gives a reasonably good result. Also, changing P* has little or no effect in summer, so you shouldn't think about the drift-concentration relationship here. Finally: Why don't you plot s_h and epsilon_h against P*? Here's an opportunity to use your new metrics to simplify the analysis!

P.12 L.11: I've already said that the anti-correlation between drift speed and thickness is (mostly) coincidental in summer, but here you can conclusively show that this is the case (at least in your model). In figure 13a you plot drift speed versus ice thickness for different P*. The figure shows that changing P* has much less effect on the drift speed in summer than winter (and no effect at all in July). Since changing P* has the same effect as changing the thickness we can conclude that the speed-thickness relationship in summer is weak to non-existing. This is something new that I'm not aware of anyone conclusively demonstrating.

P.12 .L15: Be careful when comparing to previous studies here, because the frequency (and indeed the source) of the forcing can influence the optimal choice for P*

P.14 L.1 You added a discussion about your lambda experiments, but I don't really see that this is necessary or helpful for the paper.

P.15 L.1: I like your section 4.4

P.15 L.31: Again, I'm not sure that you can't single out a value for P* that is optimal (in some sense at least).

P.16 L.5: What other processes than the drift-strength feedback are important? I'm not sure what you're trying to conclude here or on what it is based.

Figures:

Figure 1: This is not really needed, it's just a linear relationship.

Figure 2: Also not needed since I asked you to remove the discussion related to it.

Figure 5: The difference plots should show the absolute speed difference and the direction of the difference.

Figure 13a is not relevant since changing $P^*$ doesn't affect the concentration directly.

In general you have too many figures, some not necessary at all and some just uninteresting (especially if you restructure further). I barely looked at figures 4, 5, 7, 10, 11, and 12, since they are not involved in the most interesting part of the discussion.

---

## Author Response (AR2)

**Reply to Reviewers**

**Relationships between Arctic sea ice drift and strength modelled by NEMO-LIM3.6**

**Docquier *et al.* (2017), tc-2017-60**

We would like to thank the editor Dirk Notz and the two anonymous reviewers for their comments regarding our revised manuscript.

Please find below:

- **our answers to Referee #1 in blue**
- **our answers to Referee #2 in red**
- a track change version of the manuscript.

Our corresponding corrections in the revised manuscript are marked **in blue** when they refer to **Referee #1** and **in red** for **Referee #2**.

**1. Reply to Referee #1**

*1.1. General comments*

As a whole, the authors addressed the points raised by the two reviewers and the manuscript has been improved significantly. Particularly, the sensitivity experiment changing ice strength P* made the interpretation of the result clearer. The results summarized in Fig. 8, 9 and 11 - 13 provide useful information for modelers working with different sea ice models. The thorough examinations and assessments of the model results using observed sea ice data can be a good exemplar for forthcoming studies in similar topic. However, I suggest to address the following minor points before publication.

*1.1. Specific points*

- Page 1, line 13-15: I suggest to put an additional sentence to mention what we learned regarding drift - strength relation from the sensitivity experiment, otherwise readers cannot understand the necessity of the sensitivity experiment in the context of this study.

**Done.**

- Page 8, line 19-20: The use of the term "Bootstrap" is confusing. The author described that "The OSI SAF algorithm ... is a linear combination of the Bootstrap algorithm ... and the Bristol algorithm ..." (Page 5, line 23-25), while at the same time, provided a sentence "The Bootstrap algorithm provides higher concentration than OSI SAF, ..." (Page 8, line 19-20). A consistent use of the term or further explanation is needed.

**The OSI SAF algorithm is a combination of Bootstrap and Bristol algorithms, but we also use the concentration data from the Bootstrap algorithm only in our study (we call it 'Bootstrap'). We added a sentence at the end of the description of this dataset in Section 2.2.2.**

- Page 10, line 20; Page 12, line 3; Page 15, line 21; page 16, line 3: The authors used a term "interaction" to describe the relation between ice drift and ice strength, while in this study, only one direction of the dynamical influence (i.e., effect of ice strength on the drift speed) was examined and the other direction (i.e., effect of ice drift change on ice strength) was not examined. From this point of view, I think it is appropriate to use a term "relation" instead of "interaction", since the study did not clarify the consequence of ice drift change on ice strength.

**We replaced 'interactions' by 'relationships' throughout the whole manuscript (including the title) as we agree with the reviewer that 'interactions' is not really appropriate, but we think that 'relationship' is more appropriate than 'relation' as in Olason and Notz (2014).**

- Page 12, line 13-14: I think "control" is not a suitable verb to describe the results shown in this study, since it implies the drift speed is a function of ice concentration and thickness only. As shown in the hysteresis loop in Fig. 11 and 13, there are other unknown controlling factors for ice drift speed. Therefore, I suggest to rephrase ".. is controlled .. " by " .. is strongly influenced ..".

**Done.**

- Page 15, line 23-25: As mentioned above, the existence of the hysteresis loop indicates there are other unknown factors which influence ice drift speed. I suggest to address this for future study.

**Done.**

- Figure 13 a: It is difficult to identify the correspondence of the color of the regression lines to the different P* values. Please provide additional legend to describe the correspondence.

**We kept three out of the five P* curves as in panel (b) in order to enhance readability and we modified the legend to make the correspondence with the regression lines clearer.**

**2. Reply to Referee #2**

I would like to thank the authors for their responses to my previous remarks. It is clear that they took my remarks under serious consideration. They have, in particular restructured and refocused the article as, in fact both I and the other reviewer felt was necessary. As it stands, the paper is much closer to be acceptable for publication, than was the original submission.

Despite the good progress made I still have two major and some minor remarks that I would like the authors to respond to, before I can recommend publication.

*2.1. Major comments*

My first major comment is about the drift vs. thickness plots. Reading Olason and Notz, it is my understanding that they view this relationship to be only of relevance during winter, i.e. from November to March. And I agree. The reason for this is that it is only in winter that the ice cover is compact enough so that the ice thickness can play a role in the momentum equation. If the ice cover is very loose the open water part of the cover will "deform", so to speak, regardless of the thickness of the ice. The straight line between November and March is thus physical, while the straight line between May and September is coincidental.

You can also see this when looking at the IABP vs PIOMAS curve. I know I said "PIOMAS is not observations" last time, but we know that the shape of the Rothrock et al curve is wrong (they say as much in the paper themselves), and it's reasonable to assume that PIOMAS is closer to reality in that regard. In IABP vs PIOMAS the straight line in summer is only between July and September; it's still a straight line, but I would argue that this is coincidental.

Keeping this in mind your metrics should refer only to the November-March period, but it seems the entire curve is taken into account. This is crucial. Also, if you did this you could use your figure 13 b) to see that w.r.t. drift speed P* = 27.5 gives (close to) the right slope, i.e. drift speed-thickness relationship.

**We agree with the reviewer that it makes more sense that the relationship between drift speed and thickness is really physical only in winter when ice concentration is high. However, from a statistical point of view, this relationship also occurs in summer and we cannot firmly exclude any kind of relationship / causality between both variables at that time of the year.**

**Our metrics were computed over the whole year in the previous version of our manuscript. Following the very good suggestion from the reviewer, we computed our metrics separately for summer (May-September) and winter (November-March), but we prefer keeping the metrics computed over the whole year as the reference in our manuscript since the separation between 'physical' and 'non-physical' months is highly subjective. Also, taking winter months improves the results in terms of drift-thickness normalised distance (except for P* = 5.5 kN/m²) but deteriorates the results for the drift-thickness slope ratio (Fig. 10).**

**We added some text related to the methodology (Section 2.3) and the new results (Sections 3.2 and 3.3) as well as a figure showing our metrics against P* for the sensitivity experiments (Fig. 10 in the revised manuscript). In Fig. 10b, we show that P* = 27.5 kN/m² is better than P* = 5.5 and 100 kN/m² in terms of drift-thickness slope ratio for winter (solid blue curve), but P* = 45 kN/m² is the best fit (this curve is not shown in Fig. 9b [Fig. 13b in the previous version of the manuscript]).**

My second major comment is that I feel that the structure of the paper can still, relatively easily be improved. I will not stand particularly firm on this, it is more a stylistic comment than a strictly scientific one. But I honestly believe that by improving the structure you can make the paper better focused on the important new methods and results, which will help in making it more widely read, recognised, and cited.

I suggest you really focus on the new metrics you developed. You would then first introduce the metrics and then evaluate NEMO-LIM using them. This second step shows the applicability of the metrics and evaluates the model at the same time. You can then use the P* experiments to show how you can use the metrics to aid with sensitivity studies. A plot of s_h and epsilon_h against P* w. Finally you show, through the P* experiments, that the drift speed in summer doesn't depend on ice thickness.

This way you put the new metrics and new science in foreground and the evaluation of NEMO-LIM in the background. That would, in my opinion, change the paper from a somewhat utilitarian model evaluation paper to one presenting new metrics and new findings.

**We really appreciate this comment which aims at improving our paper. Following the suggestions from the reviewer, we removed some text (especially related to trends) and figures (Figs. 4, 6 and 7) from the evaluation part (Section 3.1). We computed our metrics over summer and winter months separately (see previous major comment), added some text and added a figure showing our metric against P\* for the sensitivity experiments (Fig. 10 in the revised manuscript).**

*2.2. Minor comments*

P.2 L.5: Replace "At large scale" with "To first order", or something similar. Changes in ice strength, unrelated to concentration and thickness can have a large scale effect (see Girard et al and the work that follows them).

**Done.**

P.2 L.11: You introduce the phrase "drift-strength feedback", but don't use it again in the revised paper. This sentence is not needed. Also I thought we agreed not to say "feedback" :)

**We use this terminology in Section 4.2. Although we have not found a formal mathematical expression for this feedback, we cannot exclude its potential importance and prefer to keep the terminology as it is. Please note that this terminology is only used in the Introduction and in Section 4.2.**

P.2 L.16: I would like you to say "most likely caused by reduced thickness and concentration".

**Done.**

P.2 L.32: change to "... multi-model dataset suggests that in those models thicker and more packed ..."

**Done.**

P3. L.18: Change the subsection title to something along the lines of "Model and sensitivity experiment description". Now it sounds like you'll describe the model and perform the sensitivity experiments, but you're in fact describing both.

**We changed the subsection title into 'Model and sensitivity experiments'.**

P6. L.29: You added "sea ice area", but do you really use that? Also, how does that differ from sea ice concentration. One is the true area and the other the fractional area, right?

**We use sea ice area in Fig. 2a (numbering of the revised manuscript) as suggested by the editor due to uncertainties linked to sea ice extent. In our manuscript, we define sea ice area as the total area of sea ice cover, i.e. the sum of fractional sea ice area over all grid cells. Sea ice concentration is the percentage of ocean area covered by sea ice for each grid cell separately.**

P7. L.4: You discuss the monthly mean drift speed computed from the daily components of ice velocity at length here. But I don't think it's needed. You don't use this information in the rest of the paper and it's not really relevant to what you're presenting here. Please remove this discussion.

**We think this methodological information is crucial since the values of monthly mean drift speed are twice higher if computed from daily components compared to monthly components, as demonstrated in our Fig. 1 (numbering of the revised manuscript). This was not shown previously to the best of our knowledge and the results from our model evaluation would have been different if we had used monthly components instead, so we think this merits some attention. Some previous studies (e.g. Rampal et al., 2011) compute monthly drift speed from monthly components and the agreement with observations is worse due to this.**

P.7 L.31: When introducing the metrics you don't say which months you use to calculate them. For the concentration related metrics this is probably June-October already(?), but for the thickness it should be November-March, as I mention in the major comments.

**We computed our metrics over the whole period. But we agree with the reviewer that this is also a good idea to compute them over specific time periods, so we did it and added the new results in our revised manuscript (Section 3.2). We also added a new figure (Fig. 10). Please see also our response to the first major comment above.**

P.8 L.12: the part of "which is based on a combination ... on the mean 1984-2000 seasonal cycle" is not really necessary here.

**We removed this part from the manuscript.**

P.9 L.10: You should show the OSI SAF drift, even if it doesn't cover summer. There is a huge discrepancy between the two observations you do show, so throwing in the third gives us a better idea of how serious this discrepancy is. Also, the NSIDC data is not really good in summer, last I checked.

**We added the OSI SAF drift speed in Fig. 2d. As you can see, the OSI SAF product is within the range of the IABP product (except maybe for November), which confirms the relatively low reliability of the NSIDC product in terms of mean seasonal cycle. However, please note that the OSI SAF period is much shorter (2007-2015) and does not include summer months (June-September). We added this information in the text.**

P.10 L.5: (Paragraph) See my major point about drift-thickness relationship being coincidental in summer.

**Please see our response to the first major point above.**

P10. L.13: (Paragraph) I would skip this paragraph and figure. It's not a bad idea, but you need to dedicate much more time/space to discuss the physics and observations, since you have no one to cite on this. For figure 8c: This makes sense, since you'd expect a trend in concentration to coincide with a trend in speed, but the model does the oposite (which is interesting). Why don't you show the NSIDC speeds here? For figure 8d: I can't read anything useful out of this one: A trend in thickness coincides with a trend in drift speed mainly in months where there is no relationship between drift speed and thickness.

**We removed this paragraph and Fig. 8c-d.**

P.10 L.23: Skip this last line as it refers to the paragraph I recommend skipping above.

**Done.**

P.11 L.15: (Paragraph) If you just consider the winter months (which is where changing P* really has an effect) then you see that P* = 45 kN/m^2 is actually quite a good fit.

**We already said that P\* = 45 kN/m² is a good fit from December to February in this paragraph.**

P.11 L.20: (Paragraph) If you consider only November-March for the drift-thickness relationship then P* = 27.5 kN/m^2 gives a reasonably good result. Also, changing P* has little or no effect in summer, so you shouldn't think about the drift-concentration relationship here. Finally: Why don't you plot s_h and epsilon_h against P*? Here's an opportunity to use your new metrics to simplify the analysis!

**By computing our metrics over winter months (November-March), P\* = 45 kN/m² is the best fit for the slope ratio (s_h = 0.6) and P\* = 27.5 kN/m² is the best fit for the normalised distance (epsilon_h = 4.1%) (see our new Fig. 10b).**

**We agree with the reviewer that the effect of varying P\* on drift speed is small in summer but the shape of the drift-thickness relationship is different since thickness changes. So we still think this is valuable.**

**As for the last sub-comment, we added a new figure plotting our metrics against P\* (Fig. 10). This is a very good idea and we thank the reviewer for this suggestion.**

P.12 L.11: I've already said that the anti-correlation between drift speed and thickness is (mostly) coincidental in summer, but here you can conclusively show that this is the case (at least in your model). In figure 13a you plot drift speed versus ice thickness for different P*. The figure shows that changing P* has much less effect on the drift speed in summer than winter (and no effect at all in July). Since changing P* has the same effect as changing the thickness we can conclude that the speed-thickness relationship in summer is weak to non-existing. This is something new that I'm not aware of anyone conclusively demonstrating.

**We guess that the reviewer talks about Fig. 13b, which is Fig. 9b in the revised manuscript. We agree with the reviewer that the effect of varying P\* is small in summer but there is an effect, especially when looking at P\* = 5.5 kN/m² compared to the other P\* values and when looking at August and September (Figs. 5b and 9b in the revised manuscript). We added a sentence related to this point.**

P.12 .L15: Be careful when comparing to previous studies here, because the frequency (and indeed the source) of the forcing can influence the optimal choice for P\*

**We already emphasize several potential factors that could lead to differences between our study and the one from Steele et al. (1997). We added the forcing in our list.**

P.14 L.1 You added a discussion about your lambda experiments, but I don't really see that this is necessary or helpful for the paper.

**We prefer keeping this information since it is only one paragraph and we think it provides useful information for future modelling studies.**

P.15 L.1: I like your section 4.4

**Thank you.**

P.15 L.31: Again, I'm not sure that you can't single out a value for P\* that is optimal (in some sense at least).

**We modified this conclusion due to further investigation where we find that P\* = 45 kN/m² is a good candidate compared to other P\* values in terms of the metrics used (Fig. 10) but we could not prove that this P\* value is highly superior to the others, especially when looking at the seasonal cycles of thickness and drift speed compared to observations (Fig. 5).**

P.16 L.5: What other processes than the drift-strength feedback are important? I'm not sure what you're trying to conclude here or on what it is based.

**We removed this sentence.**

Figures:

Figure 1: This is not really needed, it's just a linear relationship.

**We removed this figure.**

Figure 2: Also not needed since I asked you to remove the discussion related to it.

**Please see our response to comment P.7 L.4. We think that this information is important in the context of our study as our results are based on the computation of monthly mean drift speed. We decided to keep this figure as it clearly shows the ratio between drift speed computed from daily components and the one computed from monthly components.**

Figure 5: The difference plots should show the absolute speed difference and the direction of the difference.

**We prefer keeping the figure as it is since we think it is easier to identify regions of model overestimation (in red) and underestimation (in blue) this way.**

Figure 13a is not relevant since changing P* doesn't affect the concentration directly.

**We agree with the reviewer that changes in P\* do not affect sea ice concentration the same way as sea ice thickness. However, the shape of the drift-concentration relationship is affected since drift speed is impacted, which provides different metric values for the different P\* experiments. Therefore, we prefer keeping this sub-figure (Fig. 9a in the revised manuscript).**

In general you have too many figures, some not necessary at all and some just uninteresting (especially if you restructure further). I barely looked at figures 4, 5, 7, 10, 11, and 12, since they are not involved in the most interesting part of the discussion.

**We removed Figs. 1, 4, 6, 7 and 8c-d (previous numbering) but we kept the other ones as they still provide key information for our article. We added a new figure relating our metrics to P\* as suggested by the reviewer (Fig. 10).**

[revised manuscript text omitted]

---

## Author Response (AR3)

**Reply to Reviewers**

**Relationships between Arctic sea ice drift and strength modelled by NEMO-LIM3.6**

**Docquier *et al.* (2017), tc-2017-60**

We would like to thank the editor Dirk Notz and the reviewer for his comments regarding our revised manuscript. Please find below **our answers to Referee #2 in red**.

**1. Reply to Referee #2**

Overall the paper is well written and contains useful information for the sea ice modelling community. Particularly the thorough assessment of model results by observed data, including the diagnostics introduced in this study is a good example for modelling study. Before publication, I would suggest to address the following minor points.

- Page 9, line 11-12: "The main pattern of ice circulation, i.e., Beaufort Gyre and Transpolar Drift, are reasonably well represented by the model (Figs. 3a and 3d)". It is not possible to distinguish spatial pattern of the ice drift field from Figs. 3a and 3d, since the figures don't show vector field.

I suggest to superimpose vector arrows showing the mean drift field in Fig. 3, or revise the sentence such as "The main pattern.... (not shown)".

**Vector arrows are present in Figs. 3a and 3d. We added a sentence in the figure caption to make it clear.**

- Page 15, line 16-17: This sentence seems to me contradicting with a paragraph, which describes the limitation of the current NEMO-LIM3.6 experiments (sec. 4.3). Since this sentence does not contain any useful information, I suggest to remove. An alternative choice is to revise the sentence by merging with the next sentence, since the important information here is that the model succeeded to reproduce the relation between observed ice properties (concentration, thickness and drift speed). The capability of reproducing ice extent (or bias of ice extent) outside the analysis domain is not relevant to the conclusion of this study.

**We removed this sentence.**

[revised manuscript text omitted]